# ROME is Forged in Adversity: Robust Distilled Datasets via Information Bottleneck

**Zheng Zhou** [1]   **Wenquan Feng** [1]   **Qiaosheng Zhang** [2,3]   **Shuchang Lyu** [1]   **Qi Zhao** [1]   **Guangliang Cheng** [4]

## Abstract

Dataset Distillation (DD) compresses large datasets into smaller, synthetic subsets, enabling models trained on them to achieve performance comparable to those trained on the full data. However, these models remain vulnerable to adversarial attacks, limiting their use in safety-critical applications. While adversarial robustness has been extensively studied in related fields, research on improving DD robustness is still limited. To address this, we propose **ROME**, a novel method that enhances the adversarial **RO**bustness of DD by leveraging the Infor**M**ation Bottlen**E**ck (IB) principle. ROME includes two components: a performance-aligned term to preserve accuracy and a robustness-aligned term to improve robustness by aligning feature distributions between synthetic and perturbed images. Furthermore, we introduce the Improved Robustness Ratio (I-RR), a refined metric to better evaluate DD robustness. Extensive experiments on CIFAR-10 and CIFAR-100 demonstrate that ROME outperforms existing DD methods in adversarial robustness, achieving maximum I-RR improvements of nearly 40% under white-box attacks and nearly 35% under black-box attacks. Our code is available at https://github.com/zhouzhengqd/ROME.

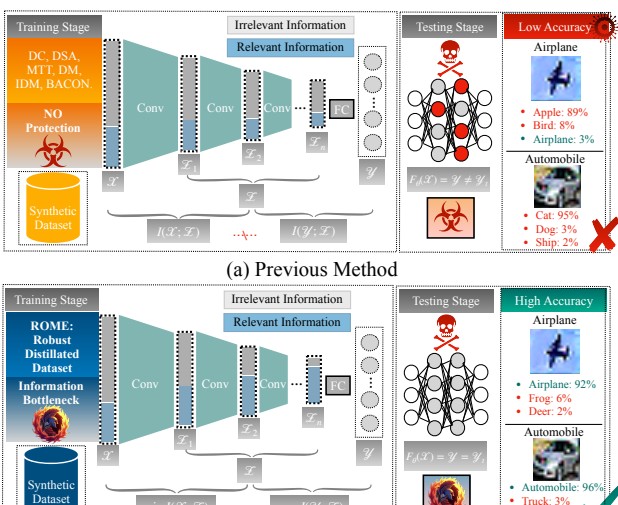

(a) Previous Method

(b) Our Method

*Figure 1.* **Comparison of previous DD methods and the proposed ROME under adversarial attacks:** (a) Previous DD methods align representations between original and synthetic datasets but remain vulnerable to adversarial attacks due to neglecting the mutual information among input $\mathcal{X}$, latent representations $\mathcal{Z}$, and output $\mathcal{Y}$, leading to reduced accuracy under perturbations. (b) Our method, ROME, employs the information bottleneck principle to minimize mutual information between $\mathcal{X}$ and $\mathcal{Z}$, while maximizing it between $\mathcal{Y}$ and $\mathcal{Z}$, thereby enhancing adversarial robustness and maintaining high accuracy under perturbations.

## 1. Introduction

The rapid expansion of large datasets has driven significant advancements in computer vision and deep learning applications, including large language models (Vaswani, 2017;

[1]Beihang University, Beijing, China [2]Shanghai Artificial Intelligence Laboratory, Shanghai, China [3]Shanghai Innovation Institute, Shanghai, China [4]University of Liverpool, Liverpool, UK. Correspondence to: Shuchang Lyu <lyushuchang@buaa.edu.cn>.

*Proceedings of the 42^{nd} International Conference on Machine Learning*, Vancouver, Canada. PMLR 267, 2025. Copyright 2025 by the author(s).

Chang et al., 2024) and large vision-language models (Radford et al., 2021), enabling models to achieve high accuracy and generalization across diverse domains (Thirunavukarasu et al., 2023; Kirchenbauer et al., 2023). However, training on such massive datasets presents substantial challenges, including high computational costs, excessive memory usage, and prolonged training times, particularly in resource-constrained environments (Lei & Tao, 2024; Yu et al., 2023).

Dataset Distillation (DD) (Wang et al., 2018) addresses these challenges by compressing large datasets into small, synthetic subsets, allowing models to achieve performance comparable to those trained on full datasets. Despite the success

of methods like DC (Zhao et al., 2021), MTT (Cazenavette et al., 2022), DM (Zhao & Bilen, 2023), IDM (Zhao et al., 2023), and BACON (Zhou et al., 2024b), models trained on distilled datasets remain vulnerable to adversarial attacks (Zhou et al., 2024a; Wu et al., 2024), as illustrated in Figure 1(a). These vulnerabilities pose significant risks to critical applications such as face recognition (Wei et al., 2022a;b), object detection (Zhou et al., 2024c; Hu et al., 2021), and autonomous driving (Wang et al., 2021a; Yuan et al., 2023), underscoring the need for robust dataset distillation techniques that enhance adversarial robustness without compromising performance.

While adversarial robustness distillation (Goldblum et al., 2020; Kuang et al., 2023; Wang et al., 2024) has been extensively explored to enhance Adversarial Robustness (AR) in knowledge distillation (Hinton, 2015; Gou et al., 2021), its application to DD remains limited. Recent efforts have made strides in addressing this gap. For instance, Wu et al. (2024) introduced DD-Robustbench, a framework for evaluating AR across distilled datasets, while Zhou et al. (2024a) proposed BEARD, a game-theoretic framework with three metrics for AR evaluation, including adversarially trained models. Additionally, Xue et al. (2024) developed GUARD, which enhances AR through curvature regularization. Despite these advances, many methods rely on adversarial training (Madry et al., 2018), which retrains models with adversarial examples to improve AR. However, applying adversarial training to DD presents two significant challenges:

  a. *High computational cost of retraining.*

  b. *Trade-off between model performance and robustness.*

To address these challenges, we propose a novel method called **ROME**, which enhances the adversarial **RO**bustness of DD using the Infor**M**ation Bottlen**E**ck (IB) (Tishby et al., 2000) principle. ROME reduces computational costs for downstream tasks while maintaining both robustness and performance. By leveraging the Conditional Entropy Bottleneck (CEB) (Fischer, 2020), a variant of IB with label priors, ROME minimizes the mutual information between input $\mathcal{X}$ and latent representations $\mathcal{Z}$, while maximizing it between output $\mathcal{Y}$ and $\mathcal{Z}$ to enhance the relevant information, as shown in Figure 1(b). This design leads to two key components: a **performance-aligned term** to preserve accuracy, and a **robustness-aligned term** to improve adversarial robustness by aligning feature distributions between synthetic and perturbed images. Additionally, we introduce a robust prior for dataset generation by applying adversarial perturbations based on the CEB theory, which can be further strengthened by incorporating robust pretrained models (Hendrycks et al., 2019; Goldblum et al., 2020). Furthermore, we propose the Improved Robustness Ratio (I-RR), an enhanced metric derived from BEARD (Zhou et al.,

2024a), to better evaluate adversarial robustness. Extensive experiments on CIFAR-10 and CIFAR-100 demonstrate that ROME outperforms existing DD methods, achieving maximum I-RR improvements of nearly 40% and nearly 35% under white-box and black-box attacks, respectively.

Our **main contributions** are summarized as follows:

- Theoretically, we *first* introduce the IB to DD for deriving robust distilled datasets. By leveraging the conditional entropy bottleneck, we obtain a numerically feasible lower bound and reformulate the IB principle, incorporating adversarial perturbations as a prior for feature learning. This approach is termed **RO**bust distilled datasets via infor**M**ation bottlen**E**ck (**ROME**).

- Algorithmically, to implement ROME, we propose two key training terms: a performance-aligned term to preserve the accuracy of models trained on distilled datasets, and a robustness-aligned term that maximizes the margin between synthetic images and the decision boundary to improve adversarial robustness.

- Experimentally, we introduce I-RR, a refined metric for more effective evaluation of DD robustness. Extensive experiments on benchmark datasets including CIFAR-10 and CIFAR-100 demonstrate that our method outperforms existing DD approaches in adversarial robustness under both white-box and black-box attacks.

## 2. Related Work

### 2.1. Dataset Distillation

Dataset distillation, introduced by Wang et al. (2018), is a bi-level optimization problem that is computationally expensive due to nested recursion. Zhao et al. (2021) proposed Dataset Condensation (DC), which improves performance by aligning gradients between original and synthetic datasets. Zhao & Bilen (2021) introduced DSA to enhance distillation via data augmentation, while Cazenavette et al. (2022) proposed MTT to match training trajectories. Zhao & Bilen (2023) introduced Distribution Matching (DM), later refined by Zhao et al. (2023) as Improved Distribution Matching (IDM). Zhou et al. (2024b) proposed BACON, a Bayesian method for DD that improves performance.

### 2.2. Adversarial Robustness Distillation

Adversarial robustness distillation, introduced by Goldblum et al. (2020), was initially proposed to improve AR in knowledge distillation and demonstrates that smaller models can achieve enhanced robustness without incurring additional training costs. Zi et al. (2021) showed that soft labels from the teacher model significantly improve the robustness of the student model. Furthermore, Kuang et al. (2023) addressed

the limitations of adversarial training (Madry et al., 2018) by proposing Information Bottleneck Distillation (IBD), which leverages the information bottleneck principle to enhance robustness. Wang et al. (2024) introduced DFARD, a method that enables robust model training without the need for additional data. Despite notable progress, research on enhancing AR in DD remains relatively scarce. Prominent works in this field include DD-Robustbench (Wu et al., 2024), a benchmark tailored for evaluating AR in distilled datasets, and BEARD (Zhou et al., 2024a), which provides a thorough AR evaluation using a game-theoretic framework and three distinct metrics. BEARD also examines models adversarially trained on distilled datasets, thereby boosting AR. However, this approach necessitates retraining and introduces a trade-off between model performance and robustness. Furthermore, Xue et al. (2024) introduced GUARD, a method that enhances AR through curvature regularization. Nonetheless, GUARD requires specific assumptions, such as the convexity of the loss function and the linearity of the feature extractor, which may restrict its applicability across diverse models and tasks.

In contrast, we propose ROME, a novel method that enhances both performance and robustness by integrating the information bottleneck principle. ROME improves AR while maintaining high accuracy under adversarial attacks, eliminating the need for retraining. Inspired by Kuang et al. (2023), ROME leverages the conditional entropy bottleneck within DD, providing a flexible and general framework that effectively balances model performance and robustness. To implement ROME, we introduce two key components: a performance-aligned term to ensure the accuracy of models trained on distilled datasets, and a robustness-aligned term that maximizes the margin between synthetic images and the decision boundary to enhance AR. Extended background and formal definitions are provided in Appendix B.

## 3. Robust Dataset Distillation via Information Bottleneck

### 3.1. Preliminary

**Motivation.** Dataset distillation compresses large datasets into compact subsets while preserving comparable performance. While adversarial robustness is well studied in related areas such as knowledge distillation (Goldblum et al., 2020), it remains underexplored in DD, which focuses on efficiency and accuracy. Adversarial training, a common approach to enhance robustness, suffers from two key limitations: *(a) high computational cost of retraining*, and *(b) trade-off between model performance and robustness*.

Adversarial examples are non-robust features (Ilyas et al., 2019) that can be mitigated through their removal. The information bottleneck principle (Tishby et al., 2000) addresses

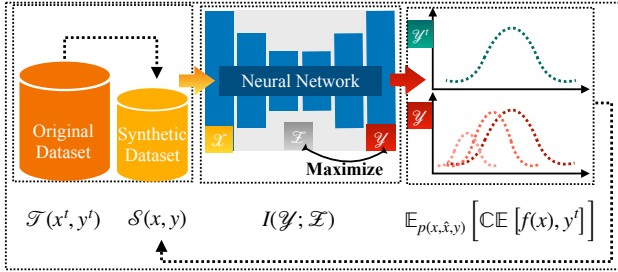

(a) Performance-aligned Term

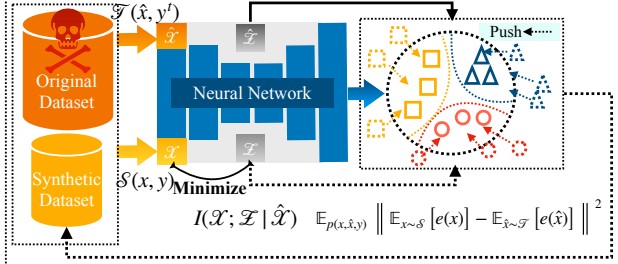

(b) Robustness-aligned Term

*Figure 2.* **The framework of ROME:** ROME utilizes the information bottleneck to frame the robust dataset distillation problem as a min-max optimization of mutual information. It consists of two key components: (a) The performance-aligned term maximizes the mutual information between the latent space $\mathcal{Z}$ and the output $\mathcal{Y}$ by aligning the logits with the true labels. (b) The robustness-aligned term minimizes the mutual information between $\mathcal{Z}$ and the input $\mathcal{X}$, conditioned on a robust prior $\hat{\mathcal{X}}$ (the adversarially perturbed dataset), by aligning the embeddings to reduce the discrepancy.

this by minimizing mutual information between input and hidden layers while maximizing it between hidden layers and output, effectively filtering irrelevant input information and enhancing adversarial robustness. Prior work (Wang et al., 2021b; Xu et al., 2022; Kuang et al., 2023) has demonstrated IB's effectiveness in improving robustness. Inspired by IBD (Kuang et al., 2023), which applies IB to knowledge distillation for robustness enhancement, we integrate IB into DD to address the challenges of adversarial training. The framework of our method is illustrated in Figure 2.

**Information Bottleneck (IB).** The information bottleneck principle, introduced by Tishby et al. (2000) for information compression, has been applied to deep learning models (Tishby & Zaslavsky, 2015). IB aims to find a representation $\mathcal{Z}$ that preserves as much information as possible about the target labels $\mathcal{Y}$, while reducing its dependence on the input $\mathcal{X}$. The IB objective is formulated as:

$$R_{IB} \equiv \max_{\mathcal{Z}} I(\mathcal{Y}; \mathcal{Z}) - \beta I(\mathcal{X}; \mathcal{Z}), \tag{1}$$

where $I$ denotes the mutual Information and $\beta$ controls the trade-off between $I(\mathcal{Y}; \mathcal{Z})$ and $I(\mathcal{X}; \mathcal{Z})$.

**Variational Information Bottleneck (VIB).** The variational information bottleneck (Alemi et al., 2017) extends the IB principle by leveraging variational inference to approximate mutual information with a tractable lower bound. The VIB objective is formulated as:

$$I(\mathcal{Y}; \mathcal{Z}) - \beta I(\mathcal{X}; \mathcal{Z})$$
$$\geq \mathbb{E}_{p(x,y)p(z|x)} \left[ \log q(y|z) - \beta \log \frac{p(z|x)}{q(z)} \right], \quad (2)$$

where $p(z|x)$ represents the latent space distribution, $q(y|z)$ approximates the true conditional distribution $p(y|z)$, and $q(z)$ is a fixed $K$-dimensional spherical Gaussian (i.e., $q(z) = \mathcal{N}(z|0, 1)$). VIB leverages deep neural networks to parameterize these distributions, enabling efficient handling of high-dimensional, continuous data like images and overcoming earlier constraints to discrete or Gaussian cases. The proof of Eq. 2 is provided in Appendix A.1.

**Conditional Entropy Bottleneck (CEB).** Building on VIB, Fischer (2020) proposed the conditional entropy bottleneck, which leverages label prior information to better approximate $q(z)$. The CEB can be formulated as:

$$I(\mathcal{Y}; \mathcal{Z}) - \beta I(\mathcal{X}; \mathcal{Z}|\mathcal{Y})$$
$$\geq \mathbb{E}_{p(x,y)p(z|x)} \left[ \log q(y|z) - \beta \log \frac{p(z|x)}{q(z|y)} \right]. \quad (3)$$

### 3.2. Information Bottleneck Perspective for Robust Dataset Distillation

**Definition 3.1** (Robust Distilled Datasets via Information Bottleneck (ROME))**.** Let $\mathcal{Y}$ represent a random variable corresponding to the output information, and $\mathcal{Z}$ represent a random variable corresponding to the latent space information in a neural network. Let $\mathcal{X}$ denote the synthetic dataset, and $\hat{\mathcal{X}}$ denote the source dataset. To introduce a robust prior for the source dataset, adversarial perturbations are applied to $\hat{\mathcal{X}}$. The parameter $\beta$ controls the relationship between $\mathcal{X}$, $\hat{\mathcal{X}}$, and $\mathcal{Z}$. The ROME can be defined as follows:

$$\text{ROME} = I(\mathcal{Y}; \mathcal{Z}) - \beta I(\mathcal{X}; \mathcal{Z}|\hat{\mathcal{X}}). \quad (4)$$

**Theorem 3.2.** *The variational lower bound of $I(\mathcal{Y}; \mathcal{Z})$ can be computed as follows (Proof in Appendix A.2):*

$$I(\mathcal{Y}; \mathcal{Z}) \geq \mathbb{E}_{p(y,z)} \left[ \log q(y|z) \right], \quad (5)$$

where $p(y, z)$ is the joint distribution of output information $\mathcal{Y}$ and latent information $\mathcal{Z}$ with $y \in \mathcal{Y}$ and $z \in \mathcal{Z}$, and $q(y|z)$ is the variational distribution of $\mathcal{Y}$ conditioned on $\mathcal{Z}$.

**Theorem 3.3.** *The variational upper bound of $I(\mathcal{X}; \mathcal{Z}|\hat{\mathcal{X}})$ can be calculated as follows (Proof in Appendix A.3):*

$$I(\mathcal{X}; \mathcal{Z}|\hat{\mathcal{X}}) \leq \mathbb{E}_{p(x,\hat{x})p(z|x,\hat{x})} \left[ \log \frac{p(z|x)}{q(z|\hat{x})} \right], \quad (6)$$

**Algorithm 1 RO**bust Dataset Distillation Infor**M**ation Bottle**N**eck (**ROME**)

---

**Require:** Perturbed data $\hat{\mathcal{X}}$, one-hot labels $y^t$, classes $c \in \{0, 1, \ldots, \mathcal{C} - 1\}$, pretrained logits $f(\cdot)$, embeddings $e(\cdot)$, learning rate $\eta$, iterations $T$.
**Ensure:** Robust distilled dataset $\mathcal{X}$.
 1: Initialize synthetic dataset $\mathcal{X}$ by randomly sampling from $\hat{\mathcal{X}}$ with corresponding class labels;
 2: **for** $t = 1$ to $T$ **do**
 3:    **for** $c = 0$ to $\mathcal{C} - 1$ **do**
 4:       Select $\mathcal{X}_c$ as the subset of $\mathcal{X}$ for class $c$, and $\hat{\mathcal{X}}_c$ as the subset of $\hat{\mathcal{X}}$ for class $c$;
 5:    **end for**
 6:    Compute the performance-aligned term $\mathcal{L}_{\text{Perf\_Alig}}$ using Eq. 10;
 7:    Compute the robustness-aligned term $\mathcal{L}_{\text{Rob\_Alig}}$ using Eq. 11;
 8:    Compute total loss $\mathcal{L}_{\text{TOTAL}}$ as in Eq. 12;
 9:    Update synthetic dataset: $\mathcal{X} \leftarrow \mathcal{X} - \eta\nabla_{\mathcal{X}}\mathcal{L}_{\text{TOTAL}}$;
10: **end for**
11: **return** robust distilled dataset $\mathcal{X}$

---

where $p(z|x)$ represents the conditional probability distribution of latent information $z \in \mathcal{Z}$ given the synthetic dataset $x \in \mathcal{X}$, and $q(z|\hat{x})$ denotes the variational distribution of $z \in \mathcal{Z}$ given the perturbed source dataset $\hat{x} \in \hat{\mathcal{X}}$.

**Theorem 3.4.** *The variational lower bound of ROME can be computed as follows (Proof in Appendix A.4):*

$$ROME = I(\mathcal{Y}; \mathcal{Z}) - \beta I(\mathcal{X}; \mathcal{Z}|\hat{\mathcal{X}})$$
$$\geq \mathbb{E}_{p(x,\hat{x},y)p(z|x,\hat{x},y)} \left[ \log q(y|z) - \beta \log \frac{p(z|x)}{q(z|\hat{x})} \right]. \quad (7)$$

*Remark* 3.5. By leveraging variational methods, we aim to refine our understanding and utilization of mutual information in the context of ROME. The variational lower bound $\mathbb{E}_{p(y,z)} \left[ \log q(y|z) \right]$ as detailed in Theorem 3.2, serves as a fundamental measure. It optimizes the expectation of $\log q(y|z)$ with respect to the distribution $q(y|z)$, where $q(y|z)$ acts as a variational approximation to the true conditional distribution $p(y|z)$. This bound provides a conservative estimate of $I(\mathcal{Y}; \mathcal{Z})$, ensuring that $q(y|z)$ effectively captures dependencies between output information $\mathcal{Y}$ and hidden variables $\mathcal{Z}$. Simultaneously, the variational upper bound $\mathbb{E}_{p(x,\hat{x})p(z|x,\hat{x})} \left[ \log \frac{p(z|x)}{q(z|\hat{x})} \right]$, as elucidated in Theorem 3.3, offers an upper bound for $I(\mathcal{X}; \mathcal{Z}|\hat{\mathcal{X}})$. It quantifies the expected log ratio between the true conditional distribution $p(z|x)$ of latent variables $\mathcal{Z}$ given the synthetic dataset $\mathcal{X}$ and the variational distribution $q(z|\hat{x})$, given the perturbed dataset $\hat{\mathcal{X}}$. These bounds help optimize variational distributions in ROME, improving the distillation of robust datasets while maintaining key information.

*Table 1.* Comparison of model robustness when trained using various DD methods with IPC settings of {1, 10, 50}, against both white-box targeted and untargeted attacks on the CIFAR-10 and CIFAR-100 datasets. Robustness evaluation metrics include RR and CREI, as well as their improved versions I-RR and I-CREI. The best results between the baseline and proposed methods are highlighted in **bold**, while the second-best results are underlined. Improvements in metrics compared to the second-best results are highlighted in red.

| Dataset | Method | Targeted Attack | | | | Untargeted Attack | | | |
|---|---|---|---|---|---|---|---|---|---|
| | | RR | CREI | I-RR | I-CREI | RR | CREI | I-RR | I-CREI |
| CIFAR-10 | Full-size | 20.42% | 24.98% | 67.24% | 48.39% | 28.33% | 25.12% | 28.82% | 25.36% |
| | DC [2020] | 30.79% | 29.35% | 88.51% | 58.21% | 31.87% | 26.70% | 56.02% | 38.78% |
| | DSA [2021] | 45.22% | 36.43% | 86.81% | 57.22% | 36.53% | 27.75% | 53.66% | 36.32% |
| | MTT [2022] | 36.00% | 32.26% | 83.95% | 56.24% | 33.30% | 26.26% | 48.34% | 33.77% |
| | DM [2023] | 46.01% | 36.01% | 85.76% | 55.89% | 34.50% | 28.32% | 56.19% | 39.16% |
| | IDM [2023] | 32.35% | 27.75% | 87.07% | 55.11% | 33.03% | 28.46% | 53.43% | 38.66% |
| | BACON [2024] | 36.83% | 33.05% | 84.37% | 56.82% | 32.87% | 27.20% | 50.49% | 36.01% |
| | **ROME** | **81.36%** | **55.28%** | **97.44%** | **63.32%** | **49.86%** | **35.05%** | **67.01%** | **43.62%** |
| | | (35.35 ↑) | (18.85 ↑) | (8.93 ↑) | (5.11 ↑) | (13.33 ↑) | (6.59 ↑) | (10.82 ↑) | (4.46 ↑) |
| CIFAR-100 | Full-size | 6.77% | 18.18% | 65.50% | 47.55% | 19.91% | 18.60% | 20.08% | 18.69% |
| | DC [2020] | 33.11% | 30.31% | 77.14% | 52.32% | 28.74% | 22.40% | 32.33% | 24.19% |
| | DSA [2021] | 43.97% | 35.01% | 72.97% | 49.51% | 28.53% | 20.40% | 33.29% | 22.77% |
| | MTT [2022] | 36.06% | 31.16% | 74.54% | 50.40% | 26.07% | 19.65% | 31.10% | 22.17% |
| | DM [2023] | 39.32% | 31.32% | 71.29% | 47.30% | 26.72% | 19.78% | 29.74% | 21.28% |
| | IDM [2023] | 34.44% | 27.16% | 74.57% | 47.23% | 26.28% | 20.36% | 30.83% | 22.63% |
| | BACON [2024] | 31.81% | 29.78% | 69.96% | 48.86% | 25.26% | 19.30% | 27.42% | 20.38% |
| | **ROME** | **103.09%** | **66.18%** | **100.65%** | **64.96%** | **44.10%** | **28.29%** | **46.24%** | **29.36%** |
| | | (59.12 ↑) | (31.17 ↑) | (23.51 ↑) | (12.64 ↑) | (15.36 ↑) | (5.89 ↑) | (12.95 ↑) | (5.17 ↑) |

### 3.3. Optimization Framework for Performance and Robustness Alignment

To achieve optimal performance and adversarial robustness, we maximize $\mathbb{E}_{p(x,\hat{x},y)p(z|x,\hat{x},y)}\left[\log q(y|z)\right]$ and minimize $\mathbb{E}_{p(x,\hat{x},y)p(z|x,\hat{x},y)}\left[\beta \log \frac{p(z|x)}{q(z|\hat{x})}\right]$. Simultaneously, maximizing the first term aims to enhance classification accuracy, while minimizing the second term helps reduce the discrepancy between the probability distributions of the synthetic dataset $\mathcal{X}$ and the perturbed source dataset $\hat{\mathcal{X}}$. For clarity, we refer to these objectives as the **performance-aligned term** and **robustness-aligned term**, respectively.

**Theorem 3.6.** *The performance-aligned term can also be expressed as follows (Proof in Appendix A.5):*

$$\mathcal{L}_{Perf\_Alig} = \mathbb{E}_{p(x,\hat{x},y)}\left[\mathbb{CE}\left[y^t, f(x)\right]\right], \qquad (8)$$

where $f(\cdot)$ is a pretrained model robust to adversarial attacks, and $f(x)$ denotes its logits output for input $x$. $y^t$ is the one-hot true label vector, and $\mathbb{CE}$ denotes cross-entropy.

**Theorem 3.7.** *The robustness-aligned term can also be expressed as the following lower bound, derived by scaling Pinsker's inequality (Proof in Appendix A.6):*

$$\mathcal{L}_{Rob\_Alig} = \mathbb{E}_{p(x,\hat{x},y)}\left\|\mathbb{E}_{x\sim\mathcal{X}}\left[e(x)\right] - \mathbb{E}_{\hat{x}\sim\hat{\mathcal{X}}}\left[e(\hat{x})\right]\right\|^2, \qquad (9)$$

where $\mathcal{X}$ and $\hat{\mathcal{X}}$ are class-aligned sample sets (i.e., $\mathcal{X}$ contains synthetic samples and $\hat{\mathcal{X}}$ perturbed original samples, both partitioned by the label $y$), $p(x,\hat{x},y)$ is the joint distribution, $e(\cdot)$ is the embedding layer output, and $\|\cdot\|^2$ denotes the squared Total Variation distance.

**Monte Carlo Approximation for ROME.** To approximate the expectations in Eq. 8 and Eq. 9, we apply Monte Carlo sampling. Specifically, for each class $c \in \mathcal{C} = \{0, 1, \ldots, \mathcal{C} - 1\}$, we draw synthetic samples $x$ and corresponding perturbed original samples $\hat{x}$ under class $c$. We then aggregate the sampled pairs across all classes with equal weighting to construct empirical estimates. The performance-aligned term is approximated as:

$$\mathcal{L}_{\text{Perf\_Alig}} = \sum_{c=0}^{\mathcal{C}-1} \frac{1}{|\mathcal{X}_c|} \sum_{x \in \mathcal{X}_c} \mathbb{CE}\left[y_c^t, f(x)\right], \qquad (10)$$

while the robustness-aligned term is estimated by

$$\mathcal{L}_{\text{Rob\_Alig}} = \sum_{c=0}^{\mathcal{C}-1} \left\|\frac{1}{|\mathcal{X}_c|} \sum_{x \in \mathcal{X}_c} e(x) - \frac{1}{|\hat{\mathcal{X}}_c|} \sum_{\hat{x} \in \hat{\mathcal{X}}_c} e(\hat{x})\right\|^2, \qquad (11)$$

where $\mathcal{X}_c$ and $\hat{\mathcal{X}}_c$ are the synthetic and perturbed sample subsets of category $c$, with sizes $|\mathcal{X}_c|$ and $|\hat{\mathcal{X}}_c|$, respectively.

*Table 2.* Comparison of model robustness measured by I-RR for various dataset distillation methods with IPC-50 under targeted and untargeted transfer-based and query-based black-box attacks on CIFAR-10. Best results are in **bold**, second-best underlined, and improvements over the second-best highlighted in red.

| Method | Targeted Attack | | Untargeted Attack | |
|---|---|---|---|---|
| | Transfer | Query | Transfer | Query |
| DC | 85.84% | 88.71% | 83.97% | 43.81% |
| DSA | 94.09% | 94.95% | 92.31% | 54.60% |
| MTT | 91.40% | 92.76% | 89.02% | 48.71% |
| DM | 92.22% | 93.86% | 90.36% | 57.53% |
| IDM | 92.17% | 94.37% | 89.22% | 63.23% |
| BACON | 92.46% | 94.67% | 89.25% | 63.26% |
| **ROME** | **99.90%** | **99.79%** | **98.44%** | **78.46%** |
| | (5.81 ↑) | (4.84 ↑) | (6.13 ↑) | (15.2 ↑) |

## 3.4. Overall Framework and Pseudocode

In summary, the overall loss function of ROME combines the terms in Eq. 10 and Eq. 11. The total loss function is defined as follows:

$$\mathcal{L}_{\text{TOTAL}} = (1 - \alpha)\mathcal{L}_{\text{Perf\_Alig}} + \alpha\mathcal{L}_{\text{Rob\_Alig}}, \qquad (12)$$

where the hyperparameter $\alpha$ serves as the weighting factor for the total loss function and is adjustable. By tuning $\alpha$, we can customize the loss function to optimize performance.

**Pseudocode Description.** The pseudocode for ROME is shown in Algorithm 1. ROME guides the distillation process to effectively enhance both performance and robustness. Inputs include the perturbed dataset $\hat{\mathcal{X}}$, one-hot true labels $y^t$ for synthetic dataset $\mathcal{X}$, classes $c \in \{0, 1, \ldots, \mathcal{C} - 1\}$, pretrained logits $f(\cdot)$, embeddings $e(\cdot)$, learning rate $\eta$, and total iterations $T$. The synthetic dataset $\mathcal{X}$ is initialized by sampling from $\hat{\mathcal{X}}$ with class labels. For each iteration $t = 1$ to $T$, class-wise subsets $\mathcal{X}_c$ and $\hat{\mathcal{X}}_c$ are selected. The performance-aligned term $\mathcal{L}_{\text{Perf\_Alig}}$ and robustness-aligned term $\mathcal{L}_{\text{Rob\_Alig}}$ are computed via Eq. 10 and Eq. 11, respectively. The total loss $\mathcal{L}_{\text{TOTAL}}$ is calculated using Eq. 12, and the synthetic dataset $\mathcal{X}$ is updated via gradient descent.

## 4. Experiments

In this section, we first outline the experimental setup and evaluation procedure in Section 4.1. We then assess the adversarial robustness and accuracy of various DD methods under both white-box and black-box attacks in Section 4.2. Section 4.3 presents a comparison of training efficiency for adversarially distilled datasets. Ablation studies of the proposed ROME method are provided in Section 4.4, followed by visualizations of the results in Section 4.5.

## 4.1. Experiment Settings

**Datasets and Baseline Methods.** To systematically evaluate our method, we conduct experiments using the BEARD (Zhou et al., 2024a) benchmark, which is specifically designed to assess the adversarial robustness of dataset distillation methods. The datasets used in our evaluation are CIFAR-10 (Krizhevsky, 2009) and CIFAR-100 (Krizhevsky, 2009). We compare the performance of our method against six baseline dataset distillation techniques: DC (Zhao et al., 2021), DSA (Zhao & Bilen, 2021), MTT (Cazenavette et al., 2022), DM (Zhao & Bilen, 2023), IDM (Zhao et al., 2023), and BACON (Zhou et al., 2024b).

**Evaluation Attack.** We evaluate the robustness of ROME against both white-box and black-box adversarial attacks. For white-box attacks, we adopt FGSM (Goodfellow et al., 2015), PGD (Madry et al., 2018), DeepFool (Moosavi-Dezfooli et al., 2016), C&W (Carlini & Wagner, 2017), and AutoAttack (Croce & Hein, 2020). For black-box attacks, we consider (1) transfer-based attacks, where adversarial examples generated from a surrogate model are used to evaluate models trained with different dataset distillation methods; and (2) query-based attacks, where adversarial examples are crafted by querying the model using methods such as Square (Andriushchenko et al., 2020) and SPSA (Uesato et al., 2018). Both targeted and untargeted scenarios are evaluated for comprehensive analysis.

To ensure a fairer evaluation, we introduce the Improved Robustness Ratio (I-RR), which refines the original Robustness Ratio (RR) from BEARD (Zhou et al., 2024a). Since RR can overestimate robustness when the Attack Success Rate ($\mathcal{ASR}$) is dominated by a strong attack, I-RR incorporates model accuracy ($\mathcal{ACC}$) to better balance robustness and performance. Additionally, we propose the Improved Comprehensive Robustness-Efficiency Index (I-CREI), combining I-RR with the Attack Efficiency Ratio (AE) from BEARD to jointly assess robustness and efficiency. These metrics reduce sensitivity to outlier attacks, providing a more stable and fair evaluation. In our experiments, I-RR is used for black-box robustness, where efficiency metrics (e.g., computation time) are unavailable, making I-CREI inapplicable. In white-box settings, I-CREI is preferred for its comprehensive assessment of robustness and efficiency. Formal definitions and further details are provided in Appendix C.1.

**Definition 4.1** (Improved Robustness Ratio (I-RR)). Given a neural network model $m \in \mathcal{M}$ and an adversarial attack function $a \in \mathcal{A}$, the I-RR is defined as:

$$\text{I-RR}(m; a) = 100 \times \left[1 - \frac{\overline{\mathcal{ASR}} \cdot \mathcal{ASR}^*}{\overline{\mathcal{ACC}}^2}\right], \qquad (13)$$

where $\overline{\mathcal{ASR}}$ is the average attack success rate, $\mathcal{ASR}^*$ is the maximum attack success rate, and $\overline{\mathcal{ACC}}$ is the average accuracy without adversarial attacks.

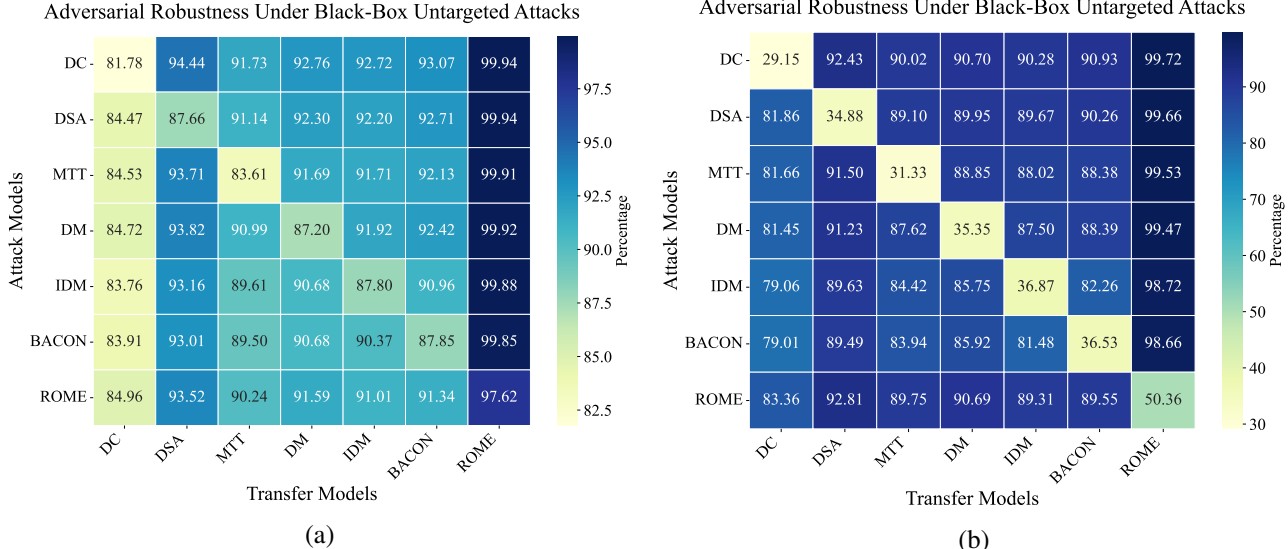

*Figure 3.* Robustness heatmap of models trained using diverse dataset distillation methods with IPC-50 on CIFAR-10 under targeted and untargeted attacks. The vertical axis represents attacked models, and the horizontal axis shows models used for transfer attacks. Heatmap values represent I-RR, with **darker colors** indicating **higher I-RR** and thus **better robustness** against adversarial attacks.

*Remark* 4.2. The I-RR metric is designed to assess adversarial robustness under a single attack $a \in \mathcal{A}$, similar to the RR introduced by BEARD (Zhou et al., 2024a). When extended to multiple adversarial attacks $\{\mathcal{A}\}$, the multi-adversary version, denoted as I-RRM, provides a more comprehensive evaluation of model robustness. Similarly, I-CREI extends I-RR by incorporating the AE to jointly assess robustness and efficiency, and can be generalized to multi-adversary settings (I-CREIM). In the following experiments, I-RR and I-CREI denote I-RRM and I-CREIM by default, as all evaluations involve multiple adversarial attacks.

**Implementation Details.** All experiments use a `ConvNet` (Gidaris & Komodakis, 2018). I-RR and AE measure the average top-1 accuracy over five runs under attacks and GPU time for attacks, respectively. The images-per-class (IPC) values are set to 50, 10, and 1, with models trained using stochastic gradient descent (SGD) with a learning rate of 0.01, momentum of 0.9, and weight decay of 0.0005. Robust priors are generated with PGD, using a perturbation budget of $\frac{8}{255}$ under targeted attack settings, and $\alpha = 0.2$ in Eq. 12, except in ablation studies. Adversarial robustness is assessed under both targeted and untargeted attacks, with a perturbation budget of $|\epsilon| = \frac{8}{255}$. For black-box attacks, the Square attack uses random search with 5000 queries, and the SPSA attack evaluates gradients with 128 random samples per iteration, both using the same budget. The setup follows BEARD guidelines, with experiments conducted on NVIDIA RTX 3090 GPUs. More experimental settings, evaluation metrics, and implementation details are provided in Appendix C.1.

### 4.2. Adversarial Robustness Evaluation

**White-box Robustness.** Table 1 shows the white-box robustness of ROME and baseline methods under targeted and untargeted attacks on CIFAR-10 and CIFAR-100. Best results are in bold, second-best are underlined, and improvements over the second-best are highlighted in red. The proposed ROME outperforms all methods in BEARD across all adversarial attacks on both datasets. Specifically, on CIFAR-10, ROME surpasses DC by 8.93% in I-RR and 5.11% in I-CREI under targeted attacks, and by 10.99% in I-RR and 4.64% in I-CREI under untargeted attacks. On CIFAR-100, ROME outperforms DC by 23.51% in I-RR and 12.64% in I-CREI under targeted attacks, and by 13.91% in I-RR and 5.17% in I-CREI under untargeted attacks. To further validate the results in Table 1, we record the accuracies of models trained on various distilled datasets under PGD attacks with perturbation budgets from 0 to 0.05, and plot the robustness curves following the protocol in (Dong et al., 2020; Liu et al., 2024). The complete curves and additional analysis are provided in Appendix C.2.

The significant improvement in adversarial robustness achieved by ROME can be attributed to the introduction of the conditional entropy bottleneck principle, which incorporates robust priors into the distilled dataset. Additionally, the use of a pre-trained robust model further strengthens these priors. Regarding the CIFAR-100 targeted attack results, where I-RR exceeds 100%, we hypothesize that this phenomenon, which we refer to as the "**Over-Robustness Phenomenon**", arises due to the application of the information bottleneck principle. During model training on ROME,

*Table 3.* Comparison of adversarial robustness (I-CREI, %) and training time (hours) of ROME and baseline dataset distillation methods on CIFAR-10 (IPC-50) under targeted attacks. "Base" indicates standard distillation training, while "+AdvTrain" refers to the additional time required for adversarial training to improve robustness. Best results, balancing robustness and efficiency, are highlighted in **bold**, and [†] denotes consistent results from "Base" to "+AdvTrain", indicating no need for adversarial fine-tuning.

| Method | I-CREI | | Training Time | |
|--------|--------|--------|--------|--------|
| | Base | +AdvTrain | Base | +AdvTrain |
| DC | 58.21% | 63.43% | 0.425 | 1.088 |
| DSA | 57.22% | 63.46% | 0.437 | 1.103 |
| MTT | 56.24% | 62.44% | 0.444 | 1.088 |
| DM | 55.89% | 63.21% | 0.452 | 1.109 |
| IDM | 55.11% | 63.11% | 0.414 | 1.055 |
| BACON | 56.82% | 62.68% | 0.442 | 1.101 |
| ROME | **63.32%** | **63.32%** [†] | **0.418** | **0.418** [†] |

both the original dataset information and non-robust features, which serve as robust priors, are effectively utilized. In the distillation process, ROME compresses substantial image information from the original dataset, leading to a higher proportion of non-robust features in the compressed dataset. Consequently, models trained with ROME may outperform clean models, exhibiting higher accuracy under adversarial attacks than in the absence of such attacks. This "**Over-Robustness Phenomenon**" explains why I-RR exceeds 100%, occurring exclusively under targeted attacks due to robust priors generated via targeted PGD.

**Black-box Robustness.** To evaluate the black-box robustness of ROME and baseline methods, we conduct comparison experiments using distilled datasets with IPC-50 on CIFAR-10. We assess performance under both targeted and untargeted transfer-based and query-based attacks. For transfer-based attacks, we perform two experiments: (1) adversarial examples generated from an adversarially trained model are transferred to evaluate the robustness of models trained with different DD methods, as shown in Table 2, and (2) adversarial examples generated from a model trained with a specific DD method are transferred to evaluate the robustness of models trained with other DD methods, as shown in Figure 3. Additional results are in Appendix C.2.

Table 2 shows the adversarial robustness measured by I-RR of ROME and baseline methods against the first transfer-based and query-based attacks. The best results between the baseline and proposed methods are highlighted in bold, while the second-best results are underlined. Improvements in metrics compared to the second-best results are highlighted in red. ROME outperforms all baseline methods in both attack types. Specifically, ROME achieves near-

*Table 4.* Ablation studies on the Robust Pretrained Model (RPM) and Adversarial Perturbation (AP) under both targeted and untargeted attacks, evaluated by I-RR and I-CREI on the CIFAR-10 dataset with IPC-50. Best results are highlighted in **bold**.

| Configuration | Targeted Attack | | Untargeted Attack | |
|--------|--------|--------|--------|--------|
| | I-RR | I-CREI | I-RR | I-CREI |
| Baseline | 81.86% | 55.26% | 32.45% | 29.29% |
| +RPM | 84.50% | 56.53% | 34.89% | 30.45% |
| +AP | 94.66% | 61.67% | 47.64% | 36.78% |
| +RPM&AP | **97.73%** | **63.23%** | **51.73%** | **38.95%** |

perfect robustness, with performance exceeding 99.79% for targeted and query-based attacks, and shows up to a 15.2% improvement in untargeted query-based attacks compared to the second-best DD method. Figure 3 presents a robustness heatmap illustrating the performance of ROME and baseline methods against the second transfer-based attack. Darker colors indicate higher robustness, while lighter colors indicate lower robustness. ROME achieves 97.62%, the best performance under targeted attacks, and 50.36%, the best performance under untargeted attacks. These results show that ROME performs strongly in both white-box and black-box attack scenarios, indicating that the robustness improvement is not due to obfuscated gradients. The consistent performance across different attack types highlights ROME's ability to generalize well, ensuring stability and effectiveness in various adversarial settings.

### 4.3. Training Efficiency Comparison with Adversarially Distilled Datasets

Table 3 presents a comparison of various DD methods in terms of adversarial robustness (measured by I-CREI under targeted attacks) and training time on CIFAR-10 with IPC-50. Among all methods, ROME achieves the highest I-CREI score (63.32%) in the "Base" setting, surpassing both traditional and recent distillation approaches. Most baselines rely on adversarial fine-tuning ("+AdvTrain") to improve robustness, often requiring significant additional training time. In contrast, ROME achieves strong robustness without adversarial retraining. Even in the "+AdvTrain" setting, ROME still outperforms the majority of methods.

In terms of efficiency, ROME requires only 0.418 hours of training in the "Base" setting, slightly more than IDM (0.414 hours) but still among the fastest methods. In the "+AdvTrain" setting, ROME incurs no additional training time, as it does not require adversarial fine-tuning. In contrast, methods like DM require 0.452 hours for base training and 0.658 hours for adversarial training, totaling over 1.1 hours. This makes ROME the most efficient method overall, offering strong robustness with the shortest total training time. Identical values in both "Base" and "+Adv-

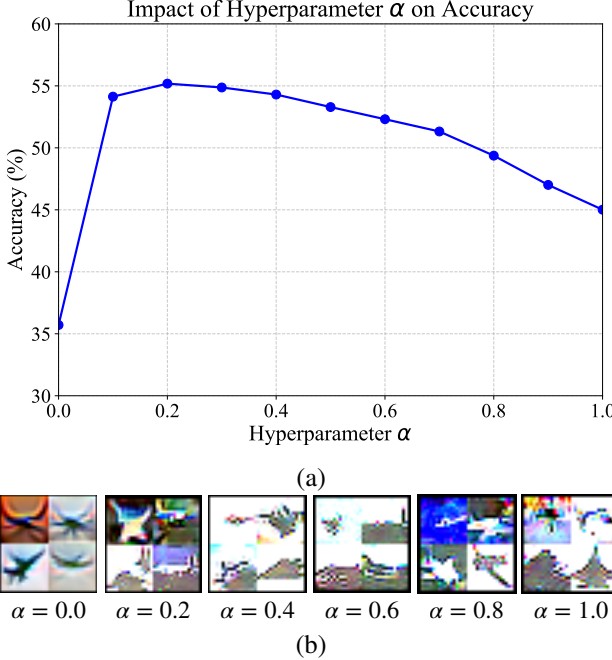

(a)

$\alpha = 0.0$ $\quad$ $\alpha = 0.2$ $\quad$ $\alpha = 0.4$ $\quad$ $\alpha = 0.6$ $\quad$ $\alpha = 0.8$ $\quad$ $\alpha = 1.0$

(b)

*Figure 4.* Ablation study of the hyperparameter $\alpha$. (a) Displays the accuracy (y-axis) as a function of $\alpha$ (x-axis) for different values of $\alpha$, and (b) shows the corresponding visualizations for these values.

Train" columns (marked by $^\dagger$) confirm no additional steps are needed. Further details on the comparison with models trained on adversarial datasets and the corresponding training time are provided in Appendix C.3.

### 4.4. Ablation Studies

**Impact of the Robust Prior.** We incorporate the robust prior into the source dataset as $\hat{x} \in \hat{\mathcal{X}}$ via Adversarial Perturbation (AP), as derived in Theorem 3.7, and further enhance it using a Robust Pretrained Model (RPM) $f(\cdot)$, as introduced in Theorem 3.6. To evaluate their contributions, we perform ablation studies on ROME under IPC-50 on CIFAR-10. As shown in Table 4, the combination of RPM and AP achieves the highest adversarial robustness, outperforming either component used in isolation. Further ablation results on the robust prior are presented in Appendix C.4, with corresponding visualizations in Appendix C.5.

**Impact of the Hyperparameter $\alpha$.** In Eq. 12, the hyperparameter $\alpha$ balances the performance-aligned and robustness-aligned terms in the loss function. As shown in Figure 4(a), varying $\alpha$ significantly affects model performance. Although the optimal accuracy on CIFAR-10 with IPC-50 is achieved at $\alpha = 0.2$, and may vary for other datasets or IPC settings (e.g., CIFAR-100; IPC-1 and IPC-10), models trained with $\alpha = 0.2$ consistently demonstrate the strongest robustness against both targeted and untargeted attacks.

### 4.5. Visualization

Figure 4(b) illustrates how varying the hyperparameter $\alpha$ affects the generated images. As $\alpha$ increases, the images become brighter and exhibit more high-frequency details, which may correspond to stronger adversarial features contributing to improved robustness. However, this increase in robustness comes with a decrease in clean accuracy (Figure 4(a)), indicating a trade-off between robustness and standard performance. Additional visualizations for ablation studies and comparisons across different IPC settings with other dataset distillation methods are in Appendix C.5.

## 5. Conclusion, Limitations, and Future Work

In this work, we introduce the information bottleneck principle into dataset distillation and propose ROME, a robust method for dataset distillation. ROME combines two key components: a performance-aligned term to preserve accuracy, and a robustness-aligned term that enhances adversarial robustness without compromising overall performance. To more effectively evaluate adversarial robustness, we introduce I-RR, a metric that balances attack success with model accuracy. Experiments on CIFAR-10 and CIFAR-100 show that ROME consistently outperforms existing methods in both white-box and black-box settings, achieving up to 38.19% higher I-RR than the worst baseline and 23.51% higher than the second-best under white-box attacks, and 34.65% and 15.2% improvements, respectively, under black-box attacks. Notably, these improvements are achieved without the need for adversarial training, significantly reducing training costs while maintaining robustness.

**Limitations and Future Work.** While ROME effectively distills robust datasets without adversarial retraining, its scalability is limited by the large search space for complex datasets like ImageNet, restricting its applicability to large-scale language and vision-language models. Future work will explore more efficient search algorithms and compression methods to scale ROME to complex tasks and datasets.

## Acknowledgements

This work was supported by the Alan Turing Institute (UK) through the project 'Turing-DSO Labs Singapore Collaboration' (SDCfP2\100009).

## Impact Statement

This paper advances machine learning security, impacting critical areas like face recognition, autonomous driving, and healthcare. It mitigates adversarial risks, promoting safer AI in high-stakes domains and supporting robust, trustworthy AI focused on fairness, accountability, and transparency.

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

# Appendix

This appendix presents supplementary details that further support the main paper, including proofs for theorems, background on dataset distillation, and experimental setup and results. The contents are organized as follows:

- Appendix A contains proofs for all theorems and definitions presented in this paper.

- Appendix B provides additional background information and preliminary details on dataset distillation.

- Appendix C provides implementation details of experiments and visualizations.

## A. Proof

### A.1. Proof of Variational Information Bottleneck (VIB)

#### A.1.1. VARIATIONAL LOWER BOUND OF VIB

We derive a variational lower bound of $I(\mathcal{Y}; \mathcal{Z})$ by leveraging the non-negativity of $\mathbb{KL}$ divergence. This bound allows us to approximate the mutual information using a tractable variational distribution $q(y|z)$.

$$I(\mathcal{Y}; \mathcal{Z}) = \int p(y, z) \log \frac{p(y, z)}{p(y)p(z)} \, dy \, dz \tag{14}$$

$$= \int p(y, z) \log \frac{p(y|z)}{p(y)} \, dy \, dz. \tag{15}$$

$$\mathbb{KL}(p(\mathcal{Y}|\mathcal{Z})\|q(\mathcal{Y}|\mathcal{Z})) \geq 0 \tag{16}$$

$$\Rightarrow \int p(y|z) \log p(y|z) \, dy \, dz \geq \int p(y|z) \log q(y|z) \, dy \, dz. \tag{17}$$

$$I(\mathcal{Y}; \mathcal{Z}) \geq \int p(y, z) \log \frac{q(y|z)}{p(y)} \, dy \, dz \tag{18}$$

$$= \int p(y, z) \log q(y|z) \, dy \, dz - \int p(y, z) \log p(y) \, dy \, dz \tag{19}$$

$$= \int p(y, z) \log q(y|z) \, dy \, dz - \int p(y) \log p(y) \, dy \tag{20}$$

$$= \int p(y, z) \log q(y|z) \, dy \, dz + H(Y) \tag{21}$$

$$\geq \int p(y, z) \log q(y|z) \, dy \, dz \tag{22}$$

$$= \int [p(x, y, z) \, dx] \log q(y|z) \, dy \, dz \tag{23}$$

$$= \int [p(x, y)p(z|x, y) \, dx] \log q(y|z) \, dy \, dz \tag{24}$$

$$= \int [p(x, y)p(z|x) \, dx] \log q(y|z) \, dy \, dz \tag{25}$$

$$= \int p(x, y)p(z|x) \log q(y|z) \, dx \, dy \, dz \tag{26}$$

$$= \mathbb{E}_{p(x,y)p(z|x)} [\log q(y|z)]. \tag{27}$$

A.1.2. VARIATIONAL UPPER BOUND OF VIB

Similarly, we provide a variational upper bound of $I(\mathcal{X}; \mathcal{Z})$ using a surrogate marginal $q(z)$, which is critical for enforcing the information bottleneck constraint in practice.

$$I(\mathcal{X}; \mathcal{Z}) = \int p(x, z) \log \frac{p(x, z)}{p(x)p(z)} \, dx \, dz \tag{28}$$

$$= \int p(x, z) \log \frac{p(z|x)}{p(z)} \, dx \, dz. \tag{29}$$

$$\mathbb{KL}(p(\mathcal{Z}) \| q(\mathcal{Z})) \geq 0 \tag{30}$$

$$\Rightarrow \int p(z) \log p(z) \, dz \geq \int p(z) \log q(z) \, dz. \tag{31}$$

$$I(\mathcal{X}; \mathcal{Z}) \leq \int p(x, z) \log \frac{p(z|x)}{q(z)} \, dx \, dz \tag{32}$$

$$= \int [p(x, z, y) \, dy] \log \frac{p(z|x)}{q(z)} \, dx \, dz \tag{33}$$

$$= \int [p(x, y)p(z|x, y) \, dy] \log \frac{p(z|x)}{q(z)} \, dx \, dz \tag{34}$$

$$= \int [p(x, y)p(z|x) \, dy] \log \frac{p(z|x)}{q(z)} \, dx \, dz \tag{35}$$

$$= \int p(x, y)p(z|x) \log \frac{p(z|x)}{q(z)} \, dx \, dy \, dz \tag{36}$$

$$= \mathbb{E}_{p(x,y)p(z|x)} \left[ \log \frac{p(z|x)}{q(z)} \right]. \tag{37}$$

Combining the above variational lower bound on $I(\mathcal{Y}; \mathcal{Z})$ and the variational upper bound on $I(\mathcal{X}; \mathcal{Z})$, we derive the following variational objective for the VIB framework:

$$I(\mathcal{Y}; \mathcal{Z}) - \beta I(\mathcal{X}; \mathcal{Z}) \geq \mathbb{E}_{p(x,y)p(z|x)} \left[ \log q(y|z) - \beta \log \frac{p(z|x)}{q(z)} \right]. \tag{38}$$

## A.2. Proof of Theorem 3.2

In the ROME setting, there are a **synthetic dataset** $\mathcal{X}$ and a **perturbed source dataset** $\hat{\mathcal{X}}$. ROME aims to learn the feature representations $\mathcal{Z}$ of $\mathcal{X}$, which is comparable to the representations of $\hat{\mathcal{X}}$. Additionally, we assume $\mathcal{Z}$, $\hat{\mathcal{X}}$ and $\mathcal{Y}$ are independent given $\mathcal{X}$, where denotes as $\mathcal{Z} \perp\!\!\!\perp \{\hat{\mathcal{X}}, \mathcal{Y}\}|\mathcal{X}$. $\mathcal{Z}$ depends only on $\mathcal{X}$ because of Markov chain $\mathcal{Z} \leftarrow \mathcal{X} \rightarrow \{\hat{\mathcal{X}}, \mathcal{Y}\}$. We can obtain the lower bound of $I(\mathcal{Y}; \mathcal{Z})$:

**Theorem A.1.** *The variational lower bound of $I(\mathcal{Y}; \mathcal{Z})$ can be computed as follows:*

$$I(\mathcal{Y}; \mathcal{Z}) \geq \mathbb{E}_{p(y,z)} \left[ \log q(y|z) \right]. \tag{39}$$

*Proof.*

$$I(\mathcal{Y};\mathcal{Z}) = H(\mathcal{Y}) - H(\mathcal{Y}|\mathcal{Z}) \propto -H(\mathcal{Y}|\mathcal{Z}) \tag{40}$$

$$= -H(\mathcal{Y}|\mathcal{Z}) \tag{41}$$

$$= \int p(y,z) \log \frac{p(y,z)}{p(z)} dy dz \tag{42}$$

$$= \int p(y,z) \log p(y|z) dy dz \tag{43}$$

$$= \int p(y,z) \log p(y|z) dy dz - \int p(y,z) \log q(y|z) dy dz + \int p(y,z) \log q(y|z) dy dz \tag{44}$$

$$= \int p(y,z) \log q(y|z) dy dz + \int \mathbb{KL}\left[p(y|z)||q(y|z)\right] dy dz \tag{45}$$

$$\geq \int p(y,z) \log q(y|z) dy dz \tag{46}$$

$$= \mathbb{E}_{p(y,z)}\left[\log q(y|z)\right]. \tag{47}$$

$\square$

### A.3. Proof of Theorem 3.3

**Theorem A.2.** $I(\mathcal{X};\mathcal{Z}|\hat{\mathcal{X}})$ *can also be simplified as follows:*

$$I(\mathcal{X};\mathcal{Z}|\hat{\mathcal{X}}) = I(\mathcal{X},\hat{\mathcal{X}};\mathcal{Z}) - I(\hat{\mathcal{X}},\mathcal{Z}) = I(\mathcal{X};\mathcal{Z}) - I(\hat{\mathcal{X}};\mathcal{Z}). \tag{48}$$

*Proof.*

$$I(\mathcal{X};\mathcal{Z}|\hat{\mathcal{X}}) = H(\mathcal{X}|\hat{\mathcal{X}}) + H(\mathcal{Z}|\hat{\mathcal{X}}) - H(\mathcal{X},\mathcal{Z}|\hat{\mathcal{X}}) \tag{49}$$

$$= H(\mathcal{Z}|\hat{\mathcal{X}}) + \left[H(\mathcal{X}|\hat{\mathcal{X}}) - H(\mathcal{X},\mathcal{Z}|\hat{\mathcal{X}})\right] \tag{50}$$

$$= H(\mathcal{Z}|\hat{\mathcal{X}}) + \left[\left(H(\mathcal{X},\hat{\mathcal{X}}) - H(\hat{\mathcal{X}})\right) - \left(H(\mathcal{X},\hat{\mathcal{X}},\mathcal{Z}) - H(\hat{\mathcal{X}})\right)\right] \tag{51}$$

$$= H(\mathcal{Z}|\hat{\mathcal{X}}) + \left[H(\mathcal{X},\hat{\mathcal{X}}) - H(\mathcal{X},\hat{\mathcal{X}},\mathcal{Z})\right] \tag{52}$$

$$= H(\mathcal{Z}|\hat{\mathcal{X}}) + \left[H(\mathcal{X},\hat{\mathcal{X}}) - \left(H(\mathcal{X},\hat{\mathcal{X}}) + H(\mathcal{Z}|\mathcal{X},\hat{\mathcal{X}})\right)\right] \tag{53}$$

$$= H(\mathcal{Z}|\hat{\mathcal{X}}) - H(\mathcal{Z}|\mathcal{X},\hat{\mathcal{X}}). \tag{54}$$

$$I(\mathcal{X},\hat{\mathcal{X}};\mathcal{Z}) - I(\hat{\mathcal{X}},\mathcal{Z}) \tag{55}$$

$$= H(\mathcal{X},\hat{\mathcal{X}}) + H(\mathcal{Z}) - H(\mathcal{X},\hat{\mathcal{X}},\mathcal{Z}) - \left[H(\hat{\mathcal{X}}) + H(\mathcal{Z}) - H(\hat{\mathcal{X}},\mathcal{Z})\right] \tag{56}$$

$$= H(\mathcal{X},\hat{\mathcal{X}}) + H(\mathcal{Z}) - \left[H(\mathcal{X},\hat{\mathcal{X}}) + H(\mathcal{Z}|\mathcal{X},\hat{\mathcal{X}})\right] - \left[H(\hat{\mathcal{X}}) + H(\mathcal{Z}) - \left(H(\hat{\mathcal{X}}) + H(\mathcal{Z}|\hat{\mathcal{X}})\right)\right] \tag{57}$$

$$= \cancel{H(\mathcal{X},\hat{\mathcal{X}})} + \cancel{H(\mathcal{Z})} - \cancel{H(\mathcal{X},\hat{\mathcal{X}})} - H(\mathcal{Z}|\mathcal{X},\hat{\mathcal{X}}) - \cancel{H(\hat{\mathcal{X}})} - \cancel{H(\mathcal{Z})} + \cancel{H(\hat{\mathcal{X}})} + H(\mathcal{Z}|\hat{\mathcal{X}}) \tag{58}$$

$$= H(\mathcal{Z}|\hat{\mathcal{X}}) - H(\mathcal{Z}|\mathcal{X},\hat{\mathcal{X}}). \tag{59}$$

$\square$

Therefore, learning intermediate features $\mathcal{Z}$ of $\mathcal{X}$ is equivalent to minimizing $I(\mathcal{X};\mathcal{Z}|\hat{\mathcal{X}})$.

$$I(\mathcal{X};\mathcal{Z}|\hat{\mathcal{X}}) = I(\mathcal{X},\hat{\mathcal{X}};\mathcal{Z}) - I(\hat{\mathcal{X}},\mathcal{Z}) = I(\mathcal{X};\mathcal{Z}) - I(\hat{\mathcal{X}};\mathcal{Z}). \tag{60}$$

**Theorem A.3.** *The variational upper bound of $I(\mathcal{X};\mathcal{Z}|\hat{\mathcal{X}})$ can be calculated as follows:*

$$I(\mathcal{X};\mathcal{Z}|\hat{\mathcal{X}}) \leq \mathbb{E}_{p(x,\hat{x})p(z|x,\hat{x})}\left[\log \frac{p(z|x)}{q(z|\hat{x})}\right]. \tag{61}$$

*Proof.*

$$I(\mathcal{X};\mathcal{Z}|\hat{\mathcal{X}}) = I(\mathcal{X};\mathcal{Z}) - I(\hat{\mathcal{X}};\mathcal{Z}) \tag{62}$$

$$= H(\mathcal{Z}) - H(\mathcal{Z}|\mathcal{X}) - \left[H(\mathcal{Z}) - H(\mathcal{Z}|\hat{\mathcal{X}})\right] \tag{63}$$

$$= -H(\mathcal{Z}|\mathcal{X}) + H(\mathcal{Z}|\hat{\mathcal{X}}) \tag{64}$$

$$= \int p(z,x) \log \frac{p(z,x)}{p(x)} dxdz - \int p(z,\hat{x}) \log \frac{p(z,\hat{x})}{p(\hat{x})} d\hat{x}dz \tag{65}$$

$$= \int p(z,x) \log p(z|x) dxdz - \int p(z,\hat{x}) \log p(z|\hat{x}) d\hat{x}dz \tag{66}$$

$$= \int p(z,x) \log p(z|x) dxdz - \int p(z,\hat{x}) \log p(z|\hat{x}) d\hat{x}dz \tag{67}$$

$$+ \int p(z,\hat{x}) \log q(z|\hat{x}) d\hat{x}dz - \int p(z,\hat{x}) \log q(z|\hat{x}) d\hat{x}dz \tag{68}$$

$$= \int p(z,x) \log p(z|x) dxdz - \int p(z,\hat{x}) \log q(z|\hat{x}) d\hat{x}dz - \int \mathbb{KL}\left[p(z|\hat{x})||q(z|\hat{x})\right] \tag{69}$$

$$\leq \int p(z,x) \log p(z|x) dxdz - \int p(z,\hat{x}) \log q(z|\hat{x}) d\hat{x}dz \tag{70}$$

$$= \int p(x)p(z|x) \log p(z|x) dxdz - \int p(\hat{x})p(z|\hat{x}) \log q(z|\hat{x}) d\hat{x}dz \tag{71}$$

$$= \int \left[p(x,\hat{x})d\hat{x}\right] p(z|x,\hat{x}) \log p(z|x,\hat{x}) dxdz - \int p(\hat{x})p(z|\hat{x}) \log q(z|\hat{x}) d\hat{x}dz \tag{72}$$

$$= \int p(x,\hat{x})p(z|x,\hat{x}) \log p(z|x) dxd\hat{x}dz - \int p(\hat{x}) \left[p(x|\hat{x})p(z|x,\hat{x})dx\right] \log q(z|\hat{x}) d\hat{x}dz \tag{73}$$

$$= \int p(x,\hat{x})p(z|x,\hat{x}) \log p(z|x) dxd\hat{x}dz - \int p(x,\hat{x})p(z|x,\hat{x}) \log q(z|\hat{x}) dxd\hat{x}dz \tag{74}$$

$$= \int p(x,\hat{x})p(z|x,\hat{x}) \log \frac{p(z|x)}{q(z|\hat{x})} dxd\hat{x}dz \tag{75}$$

$$= \mathbb{E}_{p(x,\hat{x})p(z|x,\hat{x})} \left[\log \frac{p(z|x)}{q(z|\hat{x})}\right]. \tag{76}$$

□

## A.4. Proof of Theorem 3.4

**Theorem A.4.** *The variational lower bound of ROME can be computed as follows:*

$$ROME = I(\mathcal{Y};\mathcal{Z}) - \beta I(\mathcal{X};\mathcal{Z}|\hat{\mathcal{X}}) \tag{77}$$

$$\geq \mathbb{E}_{p(x,\hat{x},y)p(z|x,\hat{x},y)} \left[\log q(y|z) - \beta \log \frac{p(z|x)}{q(z|\hat{x})}\right]. \tag{78}$$

*Proof.*

$$I(\mathcal{Y};\mathcal{Z}) = H(\mathcal{Y}) - H(\mathcal{Y}|\mathcal{Z}) \propto -H(\mathcal{Y}|\mathcal{Z}) \tag{79}$$

$$= -H(\mathcal{Y}|\mathcal{Z}) \tag{80}$$

$$= \int p(y,z) \log \frac{p(y,z)}{p(z)} dy dz \tag{81}$$

$$= \int p(y,z) \log p(y|z) dy dz \tag{82}$$

$$= \int p(y,z) \log p(y|z) dy dz - \int p(y,z) \log q(y|z) dy dz + \int p(y,z) \log q(y|z) dy dz \tag{83}$$

$$= \int p(y,z) \log q(y|z) dy dz + \int \mathbb{KL}\left[p(y|z)||q(y|z)\right] dy dz \tag{84}$$

$$= \mathbb{E}_{p(y,z)}\left[\log q(y|z)\right] \tag{85}$$

$$= \int \left[p(x,\hat{x},y,z) dx d\hat{x}\right] \log q(y|z) dy dz \tag{86}$$

$$= \int p(x,\hat{x},y) p(z|x,\hat{x},y) \log q(y|z) dx d\hat{x} dy dz \tag{87}$$

$$= \mathbb{E}_{p(x,\hat{x},y)p(z|x,\hat{x},y)}\left[\log q(y|z)\right]. \tag{88}$$

$$I(\mathcal{X};\mathcal{Z}|\hat{\mathcal{X}}) = I(\mathcal{X};\mathcal{Z}) - I(\hat{\mathcal{X}};\mathcal{Z}) \tag{89}$$

$$= H(\mathcal{Z}) - H(\mathcal{Z}|\mathcal{X}) - \left[H(\mathcal{Z}) - H(\mathcal{Z}|\hat{\mathcal{X}})\right] \tag{90}$$

$$= -H(\mathcal{Z}|\mathcal{X}) + H(\mathcal{Z}|\hat{\mathcal{X}}) \tag{91}$$

$$= \int p(z,x) \log \frac{p(z,x)}{p(x)} dx dz - \int p(z,\hat{x}) \log \frac{p(z,\hat{x})}{p(\hat{x})} d\hat{x} dz \tag{92}$$

$$= \int p(z,x) \log p(z|x) dx dz - \int p(z,\hat{x}) \log p(z|\hat{x}) d\hat{x} dz \tag{93}$$

$$= \int p(z,x) \log p(z|x) dx dz - \int p(z,\hat{x}) \log p(z|\hat{x}) d\hat{x} dz \tag{94}$$

$$+ \int p(z,\hat{x}) \log q(z|\hat{x}) d\hat{x} dz - \int p(z,\hat{x}) \log q(z|\hat{x}) d\hat{x} dz \tag{95}$$

$$= \int p(z,x) \log p(z|x) dx dz - \int p(z,\hat{x}) \log q(z|\hat{x}) d\hat{x} dz - \int \mathbb{KL}\left[p(z|\hat{x})||q(z|\hat{x})\right] \tag{96}$$

$$\leq \int p(z,x) \log p(z|x) dx dz - \int p(z,\hat{x}) \log q(z|\hat{x}) d\hat{x} dz \tag{97}$$

$$= \int p(x)p(z|x) \log p(z|x) dx dz - \int p(\hat{x})p(z|\hat{x}) \log q(z|\hat{x}) d\hat{x} dz \tag{98}$$

$$= \int \left[p(x,\hat{x}) d\hat{x}\right] p(z|x,\hat{x}) \log p(z|x,\hat{x}) dx dz - \int p(\hat{x})p(z|\hat{x}) \log q(z|\hat{x}) d\hat{x} dz \tag{99}$$

$$= \int p(x,\hat{x})p(z|x,\hat{x}) \log p(z|x) dx d\hat{x} dz - \int p(\hat{x}) \left[p(x|\hat{x})p(z|x,\hat{x}) dx\right] \log q(z|\hat{x}) d\hat{x} dz \tag{100}$$

$$= \int p(x,\hat{x})p(z|x,\hat{x}) \log p(z|x) dx d\hat{x} dz - \int p(x,\hat{x})p(z|x,\hat{x}) \log q(z|\hat{x}) dx d\hat{x} dz \tag{101}$$

$$= \int p(x,\hat{x})p(z|x,\hat{x}) \log \frac{p(z|x)}{q(z|\hat{x})} dx d\hat{x} dz \tag{102}$$

$$= \int \left[p(x,\hat{x},y) dy\right] p(z|x,\hat{x},y) \log \frac{p(z|x)}{q(z|\hat{x})} dx d\hat{x} dz \tag{103}$$

$$= \int p(x,\hat{x},y)p(z|x,\hat{x},y) \log \frac{p(z|x)}{q(z|\hat{x})} dx d\hat{x} dy dz \tag{104}$$

$$= \mathbb{E}_{p(x,\hat{x},y)p(z|x,\hat{x},y)}\left[\log \frac{p(z|x)}{q(z|\hat{x})}\right]. \tag{105}$$

□

Therefore, the lower bound of ROME can be defined as:

$$ROME = I(\mathcal{Y}; \mathcal{Z}) - \beta I(\mathcal{X}; \mathcal{Z}|\hat{\mathcal{X}}) \tag{106}$$

$$\geq \mathbb{E}_{p(x,\hat{x},y)p(z|x,\hat{x},y)} \left[ \log q(y|z) - \beta \log \frac{p(z|x)}{q(z|\hat{x})} \right]. \tag{107}$$

*Remark* A.5. By leveraging variational methods, we aim to refine our understanding and utilization of mutual information in the context of ROME. The variational lower bound $\mathbb{E}_{p(y,z)} [\log q(y|z)]$ as detailed in Theorem 3.2, serves as a fundamental measure. It optimizes the expectation of $\log q(y|z)$ with respect to the distribution $q(y|z)$, where $q(y|z)$ acts as a variational approximation to the true conditional distribution $p(y|z)$. This bound provides a conservative estimate of $I(\mathcal{Y}; \mathcal{Z})$, ensuring that $q(y|z)$ effectively captures dependencies between output information $\mathcal{Y}$ and hidden variables $\mathcal{Z}$. Simultaneously, the variational upper bound $\mathbb{E}_{p(x,\hat{x})p(z|x,\hat{x})} \left[ \log \frac{p(z|x)}{q(z|\hat{x})} \right]$, as elucidated in Theorem 3.3, offers an upper bound for $I(\mathcal{X}; \mathcal{Z}|\hat{\mathcal{X}})$. It quantifies the expected log ratio between the true conditional distribution $p(z|x)$ of latent variables $\mathcal{Z}$ given the synthetic dataset $\mathcal{X}$ and the variational distribution $q(z|\hat{x})$, given the perturbed dataset $\hat{\mathcal{X}}$. These bounds help optimize variational distributions in ROME, improving the distillation of robust datasets while maintaining key information.

### A.5. Proof of Theorem 3.6

**Theorem A.6.** *The performance-aligned term can also be expressed as follows:*

$$\mathcal{L}_{Perf\_Alig} = \mathbb{E}_{p(x,\hat{x},y)} \left[ \mathbb{CE} \left[ y^t, f(x) \right] \right], \tag{108}$$

where $f(\cdot)$ is a pretrained model robust to adversarial attacks, and $f(x)$ denotes its logits output for input $x$. $y^t$ is the one-hot true label vector, and $\mathbb{CE}$ denotes cross-entropy.

*Proof.*

$$\mathbb{E}_{p(x,\hat{x},y)p(z|x,\hat{x},y)} [\log q(y|z)] = \mathbb{E}_{p(x,\hat{x},y)p(z|x)} [\log q(y|z)] \tag{109}$$

$$= \mathbb{E}_{p(x,\hat{x},y)} \mathbb{E}_{p(z|x)} [\log q(y|z)] \tag{110}$$

$$= \mathbb{E}_{p(x,\hat{x},y)} [\log q(y|e(x))] \tag{111}$$

$$= -\mathbb{E}_{p(x,\hat{x},y)} \left[ -y^t \log q(y^t|e(x)) \right] \tag{112}$$

$$= -\mathbb{E}_{p(x,\hat{x},y)} \left[ \mathbb{CE}[y^t, f(x)] \right]. \tag{113}$$

□

To maximize the performance-aligned term, we can equivalently minimize the negative of this term. To facilitate optimization using gradient descent, we formulate the term as follows:

$$\mathcal{L}_{\text{Perf\_Alig}} = \mathbb{E}_{p(x,\hat{x},y)} \left[ \mathbb{CE} \left[ y^t, f(x) \right] \right]. \tag{114}$$

### A.6. Proof of Theorem 3.7

**Definition A.7** (Total Variation Distance)**.** The total variation distance can be defined as follows:

$$\mathbb{TV}(P, Q) = \frac{1}{2} \sum_x |P(x) - Q(x)|. \tag{115}$$

**Definition A.8** (Kullback-Leibler Divergence)**.** The Kullback-Leibler divergence can be defined as follows:

$$\mathbb{KL}(P||Q) = \sum_x P(x) \log \left( \frac{P(x)}{Q(x)} \right). \tag{116}$$

**Definition A.9** (Pinsker's Inequality). The Pinsker's inequality can be defined as follows:

$$\mathbb{TV}(P,Q) \leq \sqrt{\frac{1}{2}\mathbb{KL}(P||Q)}. \tag{117}$$

**Theorem A.10.** *The robustness-aligned term can also be expressed as the following lower bound, derived by scaling Pinsker's inequality:*

$$\mathcal{L}_{Rob\_Alig} = \mathbb{E}_{p(x,\hat{x},y)}\|\mathbb{E}_{x\sim\mathcal{X}}\left[e(x)\right] - \mathbb{E}_{\hat{x}\sim\hat{\mathcal{X}}}\left[e(\hat{x})\right]\|^2, \tag{118}$$

where $\mathcal{X}$ and $\hat{\mathcal{X}}$ are class-aligned sample sets (i.e., $\mathcal{X}$ contains synthetic samples and $\hat{\mathcal{X}}$ perturbed original samples, both partitioned by the label $y$), $p(x,\hat{x},y)$ is the joint distribution, $e(\cdot)$ is the embedding layer output, and $\|\cdot\|^2$ denotes the squared Total Variation distance.

*Proof.*

$$\mathbb{E}_{p(x,\hat{x},y)p(z|x,\hat{x},y)}\left[\log\frac{p(z|x)}{q(z|\hat{x})}\right] = \mathbb{E}_{p(x,\hat{x},y)p(z|x)}\left[\log\frac{p(z|x)}{q(z|\hat{x})}\right] \tag{119}$$

$$= \mathbb{E}_{p(x,\hat{x},y)}\mathbb{E}_{p(z|x)}\left[\log\frac{p(z|x)}{q(z|\hat{x})}\right] \tag{120}$$

$$= \mathbb{E}_{p(x,\hat{x},y)}\mathbb{KL}\left[p(z|x)||q(z|\hat{x})\right] \tag{121}$$

$$\geq \underbrace{\mathbb{E}_{p(x,\hat{x},y)}\left[2\mathbb{TV}^2\left[p(z|x),q(z|\hat{x})\right]\right]}_{\text{Pinsker's inequality}} \tag{122}$$

$$= \mathbb{E}_{p(x,\hat{x},y)}\|\mathbb{E}_{x\sim\mathcal{X}}\left[e(x)\right] - \mathbb{E}_{\hat{x}\sim\hat{\mathcal{X}}}\left[e(\hat{x})\right]\|^2. \tag{123}$$

$$\square$$

Therefore, the robustness-aligned term can be also computed as:

$$\mathcal{L}_{\text{Rob\_Alig}} = \mathbb{E}_{p(x,\hat{x},y)}\|\mathbb{E}_{x\sim\mathcal{X}}\left[e(x)\right] - \mathbb{E}_{\hat{x}\sim\hat{\mathcal{X}}}\left[e(\hat{x})\right]\|^2. \tag{124}$$

## B. Extended Background

### B.1. Dataset Distillation

Let $\mathcal{T} = \{(x_i, y_i)\}_{i=1}^{|\mathcal{T}|}$ denote the real dataset, where each $x_i \in \mathcal{X} \subset \mathbb{R}^d$ is a $d$-dimensional input and $y_i \in \mathcal{Y} = \{0, \ldots, C-1\}$ is the corresponding class label. The distilled (synthetic) dataset is represented as $\mathcal{S} = \{(\tilde{x}_i, \tilde{y}_i)\}_{i=1}^{|\mathcal{S}|}$, where $\tilde{x}_i \in \mathbb{R}^d$, $\tilde{y}_i \in \mathcal{Y}$, and $|\mathcal{S}| \ll |\mathcal{T}|$. In this work, we adopt the notation $\mathcal{X} \equiv \mathcal{S}$ for the synthetic dataset, and use $\hat{\mathcal{X}}$ to denote the adversarially perturbed version of $\mathcal{T}$. This deviates from the conventional dataset distillation literature, where $\mathcal{X}$ typically refers to the clean source data. The modified definitions are introduced to align with our focus on adversarial robustness, an aspect that has been underexplored in DD research. Notations in cited methods are retained for clarity and consistency.

The goal of dataset distillation is to construct a compact dataset $\mathcal{S}$ that captures the essential information in $\mathcal{T}$ such that training a model on $\mathcal{S}$ yields similar performance to training on $\mathcal{T}$. Let $\phi_\theta : x \mapsto y$ denote a model parameterized by $\theta$, and let $\mathcal{L}$ be a loss function (e.g., cross-entropy). The distillation objective is formulated as:

$$\mathbb{E}_{x\sim\mathcal{T}}[\mathcal{L}(\phi_\theta(x), y)] \simeq \mathbb{E}_{x\sim\mathcal{S}}[\mathcal{L}(\phi_\theta(\tilde{x}), \tilde{y})]. \tag{125}$$

#### B.1.1. META-LEARNING BASED METHOD

A predominant approach to dataset distillation formulates the task as a bi-level meta-optimization problem (Wang et al., 2018). The inner optimization minimizes the training loss on the synthetic dataset $\mathcal{S}$ to obtain model parameters $\theta(\mathcal{S})$, while the outer optimization updates $\mathcal{S}$ to minimize the generalization loss on the original dataset $\mathcal{T}$:

$$\mathcal{S}^* = \arg\min_{\mathcal{S}} \mathbb{E}_{\theta\sim\Theta}\left[l(\mathcal{T};\theta^*(\mathcal{S}))\right] \quad \text{s.t.} \quad \theta^*(\mathcal{S}) = \arg\min_{\theta} \mathcal{L}(\mathcal{S},\theta), \tag{126}$$

where $\mathcal{L}$ denotes the task-specific loss function. Despite its effectiveness, this bi-level framework introduces substantial computational overhead, spurring research into more tractable alternatives.

### B.1.2. GRADIENT MATCHING BASED METHODS

To mitigate the memory and time overhead introduced by gradient unrolling in meta-learning, Zhao et al. (2021) propose a gradient matching approach. This method optimizes the synthetic data $\mathcal{S}$ by aligning its gradient with that of the original data $\mathcal{T}$ using cosine similarity. At each iteration, class-wise mini-batches $\mathcal{S}c \subset \mathcal{S}$ and $\mathcal{T}c \subset \mathcal{T}$ are sampled, where $c \in \mathcal{C}$ indexes the classes, and updates are performed independently for each class. The objective is given by:

$$\min_{\mathcal{S}} \mathbb{E}_{\theta \sim \Theta} \left[ \sum_{c=0}^{C-1} \mathcal{D}(\nabla \mathcal{L}(\mathcal{S}_c; \theta), \nabla \mathcal{L}(\mathcal{T}_c; \theta)) \right], \tag{127}$$

where $\mathcal{D}(\cdot, \cdot)$ denotes the cosine distance between gradients. While this formulation avoids full backpropagation through time, it remains computationally intensive due to repeated gradient computations and per-class updates.

### B.1.3. DISTRIBUTION MATCHING BASED METHODS

To further reduce the computational burden of bi-level optimization, Zhao & Bilen (2023) introduce a distribution matching approach based on the Maximum Mean Discrepancy (MMD) metric. This method minimizes the Euclidean distance between the feature distributions of synthetic data $\mathcal{S}$ and real data $\mathcal{T}$ under a given model $\phi_\theta$. The objective is defined as:

$$\min_{\mathcal{S}} \mathbb{E}_{\theta \sim \Theta} \| \frac{1}{|\mathcal{S}|} \sum_{i=1}^{|\mathcal{S}|} \phi_\theta(\tilde{x}_i) - \frac{1}{|\mathcal{T}|} \sum_{j=1}^{|\mathcal{T}|} \phi_\theta(x_j) \|^2, \tag{128}$$

where $\phi_\theta(\tilde{x})$ and $\phi_\theta(x)$ denote the feature embeddings of synthetic and real samples, respectively. By aligning the feature distributions, this approach bypasses gradient-based unrolling and offers improved efficiency.

## B.2. Adversarial Robustness Distillation

### B.2.1. ADVERSARIAL TRAINING

Deep neural networks have been found to be vulnerable to adversarial examples, which are generated by adding imperceptible adversarial perturbations (Szegedy et al., 2014). Goodfellow et al. (2015) proposed the Fast Gradient Sign Method (FGSM) adversarial attack method, which generates adversarial perturbation in the direction of the gradient of the loss function. Madry et al. (2018) presented a Projected Gradient Decent (PGD) adversarial attack method, which is a multi-step optimal first-order attack method. Following these works, a series of works have been proposed to improve the performance of the adversarial attack, such as DeepFool (Moosavi-Dezfooli et al., 2016), C&W (Carlini & Wagner, 2017), AutoAttack (Croce & Hein, 2020). To overcome the effect of adversarial examples on deep neural networks, Madry et al. (2018) proposed adversarial training by adopting a multistep adversarial attack, i.e., PGD, to generate adversarial examples to improve adversarial robustness. adversarial training has been demonstrated as one of the most effective approaches to improve adversarial robustness, which can be formulated as a mini-max optimization problem as follows:

$$\arg \min_{\theta} \mathbb{E}_{(x,y) \sim \mathcal{D}} \left[ \max_{\epsilon} \mathcal{L}(f_\theta(x + \epsilon), y) \right] \quad \text{s.t.} \quad \|\epsilon\|_p \leq c, \tag{129}$$

where $f_\theta(\cdot)$ represents DNN with wights $\theta$, $\mathcal{D}$ is a data distribution with the benign sample $x$ and the correpording ground truth label $y$, $\epsilon$ is the generated adversarial perturbation, $c$ is the maximum perturbation strength and $\mathcal{L}(f_\theta(x + \epsilon), y)$ represents the cross entropy loss function. The adversarial perturbation can be defined as:

$$\epsilon_t = \prod_{[-c,c]} \left[ \epsilon_{t-1} + \alpha \, \text{sign}(\nabla_x \mathcal{L}(f_\theta(x + \epsilon_{t-1}), y)) \right], \tag{130}$$

where $\epsilon_t$ is the adversarial perturbation for $t$ iterations and $\alpha$ is the step size.

### B.2.2. INFORMATION BOTTLENECK

Information Bottleneck (IB) (Tishby et al., 2000; Alemi et al., 2017) aims to encode the maximally informative representation $\mathcal{Z}$ for target labels $\mathcal{Y}$ while restrain input informmation $\mathcal{X}$. The objective function of IB can be dedined as:

$$R_{IB} \equiv \max_{\mathcal{Z}} I(\mathcal{Y}; \mathcal{Z}) - \beta I(\mathcal{X}; \mathcal{Z}), \tag{131}$$

where $I$ denotes the mutual Information and $\beta$ controls the trade-off between $(\mathcal{Y}; \mathcal{Z})$ and $I(\mathcal{X}; \mathcal{Z})$.

# C. Experiment

## C.1. Experiment Settings

**Datasets.**   In our experiments, we use two standard image classification datasets: CIFAR-10 (Krizhevsky, 2009) and CIFAR-100 (Krizhevsky, 2009). Each dataset has been selected for its relevance and complexity in the context of dataset distillation and adversarial robustness evaluation.

- **CIFAR-10** (Krizhevsky, 2009) contains 60,000 $32 \times 32$ color images in 10 classes, with 50,000 images for training and 10,000 for testing. The images are preprocessed to normalize pixel values to the range [0, 1].

- **CIFAR-100** (Krizhevsky, 2009) Similar to CIFAR-10 but with 100 classes, this dataset contains 60,000 images, divided into 50,000 training and 10,000 testing images. Each image is resized to $32 \times 32$ pixels and normalized.

**Dataset Distillation Methods.**   We use six representative dataset distillation methods as baselines: DC (Zhao et al., 2021), DSA (Zhao & Bilen, 2021), DM (Zhao & Bilen, 2023), MTT (Cazenavette et al., 2022), IDM (Zhao et al., 2023), and BACON (Zhou et al., 2024b). These methods span several common categories in recent dataset distillation research, including gradient matching methods (Zhao et al., 2021; Zhao & Bilen, 2021), distribution matching methods (Zhao & Bilen, 2023; Zhao et al., 2023; Zhou et al., 2024b), and trajectory matching methods (Cazenavette et al., 2022).

- **DC** (Zhao et al., 2021) formulates dataset distillation as a bi-level optimization problem, focusing on matching the gradients of deep neural networks trained on the original dataset $\mathcal{T}$ and the synthetic dataset $\mathcal{S}$.

- **DSA** (Zhao & Bilen, 2021) improves distillation by incorporating data augmentation, enabling the generation of more informative synthetic images, which enhances the performance of models trained with these augmentations.

- **DM** (Zhao & Bilen, 2023) offers a straightforward yet impactful method for generating condensed images by aligning the feature distributions of synthetic images $\mathcal{S}$ with those of the original training set $\mathcal{T}$ across multiple sampled embedding spaces.

- **MTT** (Cazenavette et al., 2022) introduces trajectory matching as a distillation technique, condensing large datasets into smaller ones by aligning the training trajectories of models trained on both the synthetic $\mathcal{S}$ and original $\mathcal{T}$ datasets.

- **IDM** (Zhao et al., 2023) proposes a novel dataset condensation approach based on distribution matching, which proves to be both efficient and promising for dataset distillation tasks.

- **BACON** (Zhou et al., 2024b) leverages a Bayesian framework for dataset distillation, formulating it as risk minimization to substantially improve performance and efficiency.

**Adversarial Attack Methods.**   All attacks are implemented using the BEARD framework (Zhou et al., 2024a), which provides a comprehensive suite of state-of-the-art adversarial attack methods, including both white-box and black-box attacks. To ensure fair comparisons, consistent parameters are applied across all models. The attack library includes FGSM (Goodfellow et al., 2015), PGD (Madry et al., 2018), C&W (Carlini & Wagner, 2017), DeepFool (Moosavi-Dezfooli et al., 2016), and AutoAttack (Croce & Hein, 2020) for white-box attacks, as well as Square (Andriushchenko et al., 2020) and SPSA (Uesato et al., 2018) for black-box attacks. During evaluation, adversarial perturbations are applied to assess the robustness of distilled datasets generated by various methods. Both targeted and non-targeted attacks are conducted to comprehensively evaluate adversarial robustness. For consistency, all trained models are tested under the same parameters, with the perturbation budget set to $|\epsilon| = \frac{8}{255}$ for all methods except DeepFool and C&W.

- **FGSM** (Goodfellow et al., 2015) generates adversarial examples by perturbing the input in the direction of the gradient of the loss function, with a perturbation size set to $\epsilon = 8/255$.

- **PGD** (Madry et al., 2018) extends FGSM by applying iterative steps to create adversarial examples. The perturbation budget and step size are adjusted for each dataset to enhance attack strength.

- **C&W** (Carlini & Wagner, 2017) focuses on optimizing adversarial examples to minimize perturbation while ensuring misclassification, providing a robust evaluation of model robustness.

- **DeepFool** (Moosavi-Dezfooli et al., 2016) estimates the minimal perturbation required to induce misclassification, offering insights into the model's sensitivity to adversarial changes.

- **AutoAttack** (Croce & Hein, 2020) combines multiple strong attacks to provide a comprehensive evaluation of model robustness, ensuring thorough assessment of adversarial robustness.

- **Square** (Andriushchenko et al., 2020) utilizes a random search strategy without relying on gradient approximation. It is configured with multiple queries and a fixed perturbation budget.

- **SPSA** (Uesato et al., 2018) estimates gradients by drawing random samples and performs multiple iterations, each constrained by a fixed perturbation budget.

**Evaluation Metric.**

**Definition C.1** (Average Attack Success Rate). Let $m \in \mathcal{M}$ represent a neural network model and $a \in \mathcal{A}$ an adversarial attack function. The average attack success rate is defined to evaluate the average-case attack scenario as follows:

$$\overline{\mathcal{ASR}} = \mathbb{E}_{m \in \mathcal{M}} \mathbb{E}_{a \in \mathcal{A}} \mathcal{ASR}(m; a). \tag{132}$$

**Definition C.2** (Maximum Attack Success Rate). Given a neural network model $m \in \mathcal{M}$ and an adversarial attack function $a \in \mathcal{A}$. The maximum attack success rate is defined to capture the worst-case attack scenario as follows:

$$\mathcal{ASR}^* = \max_{m \in \mathcal{M}, a \in \mathcal{A}} \mathcal{ASR}(m; a). \tag{133}$$

**Definition C.3** (Average Accuracy). Let $m \in \mathcal{M}$ represent a neural network model and $a \in \mathcal{A}$ an adversarial attack function. The average accuracy is defined to normalize both the average- and worst-case attack conditions as follows:

$$\overline{\mathcal{ACC}} = \mathbb{E}_{m \in \mathcal{M}} \mathbb{E}_{a \in \emptyset} \mathcal{ACC}(m; a). \tag{134}$$

**Definition C.4** (Improved Robustness Ratio (I-RR)). Given a neural network model $m \in \mathcal{M}$ and an adversarial attack function $a \in \mathcal{A}$, the improved robustness ratio is defined as:

$$\text{I-RR}(m; a) = 100 \times \left[ 1 - \frac{\overline{\mathcal{ASR}} \cdot \mathcal{ASR}^*}{\overline{\mathcal{ACC}}^2} \right], \tag{135}$$

where $\overline{\mathcal{ASR}}$ is the average attack success rate, $\mathcal{ASR}^*$ is the maximum attack success rate, and $\overline{\mathcal{ACC}}$ is the average accuracy without adversarial attacks.

*Remark* C.5. The I-RR metric is designed to assess adversarial robustness under a single attack $a \in \mathcal{A}$, similar to the RR introduced by BEARD (Zhou et al., 2024a). When extended to multiple adversarial attacks $\{\mathcal{A}\}$, the multi-adversary version, denoted as I-RRM, provides a more comprehensive evaluation of model robustness. Similarly, I-CREI extends I-RR by incorporating the AE to jointly assess robustness and efficiency, and can be generalized to multi-adversary settings (I-CREIM). In the following experiments, I-RR and I-CREI denote I-RRM and I-CREIM by default, as all evaluations involve multiple adversarial attacks.

**Motivation for Introducing I-RR and I-CREI.** The Robustness Ratio (RR) (Zhou et al., 2024a) evaluates adversarial robustness by calculating the relative difference between the average ($\overline{\mathcal{ASR}}$) and worst-case ($\mathcal{ASR}^*$) attack success rates, aiming to quantify the discrepancy between these values as follows:

$$\text{RR}(m; a) = 100 \times \left[ 1 - \frac{\overline{\mathcal{ASR}}}{\mathcal{ASR}^*} \right]. \tag{136}$$

However, RR may overestimate robustness when a dominant attack disproportionately inflates $\mathcal{ASR}^*$. The Improved Robustness Ratio (I-RR) addresses this issue by ensuring that both the average ($\overline{\mathcal{ASR}}$) and worst-case ($\mathcal{ASR}^*$) attack success rates are minimized, while maintaining high clean accuracy ($\overline{\mathcal{ACC}}$). Furthermore, I-CREI extends the Comprehensive Robustness-Efficiency Index (CREI) by replacing RR with I-RR, providing a more reliable assessment that balances both attack effectiveness and computational efficiency. Together, I-RR and I-CREI establish a more robust and fair evaluation framework, particularly in scenarios where adversarial attacks vary widely in strength or dominate specific cases.

**Black-box Evaluations.** We conduct two types of transfer-based attacks and one query-based attack to evaluate the robustness of DD methods in black-box settings:

- **Transfer-based Attack from Adversarially Trained Models**: Adversarial examples are generated from an adversarially trained model and transferred to evaluate the robustness of models trained with various DD methods. This simulates an attacker who cannot access the distilled model but can use adversarial examples from a different model. Results are shown in Table 2.

- **Transfer-based Attack across Distilled Models**: Adversarial examples are generated from a model trained with a specific DD method and transferred to models trained with other DD methods. This tests the transferability of adversarial perturbations and investigates the impact of DD method choice on robustness. Results are shown in Figure 4.

- **Query-based Attack**: Adversarial examples are generated using Square (Andriushchenko et al., 2020) and SPSA (Uesato et al., 2018) through multiple rounds of querying, assessing model robustness under query-based attacks. Results are shown in Table 2.

**Implementation Details.** We use a `ConvNet` architecture (Gidaris & Komodakis, 2018) for the dataset distillation experiments. The performance of the synthetic datasets is evaluated by I-RR and AE, representing the average top-1 accuracy over five runs and the average GPU time required for adversarial attacks on the validation set, respectively. The experiments are conducted with different IPC values, specifically IPC-50, IPC-10, and IPC-1. Models are trained using the SGD optimizer with a learning rate of 0.01, momentum of 0.9, and weight decay of 0.0005. All robust priors are generated using PGD with a perturbation budget of $\frac{8}{255}$ under targeted attack settings. In our experiments, we set $\alpha$ in Eq. 12 to 0.2, except in the ablation studies. Both targeted and untargeted attacks are used to evaluate adversarial robustness. To maintain consistency, all models are trained with identical parameters, and a perturbation budget of $|\epsilon| = \frac{8}{255}$ is applied to all methods except DeepFool (Moosavi-Dezfooli et al., 2016) and C&W (Carlini & Wagner, 2017). For black-box attacks, the Square attack employs random search without gradient approximation, configured with a maximum of 5000 queries and a perturbation budget of $\frac{8}{255}$. Similarly, the SPSA attack conducts full gradient evaluations by drawing 128 random samples per iteration, utilizing a perturbation budget of $\frac{8}{255}$ in a single iteration. For fair comparisons in generalization, we incorporate DSA (Zhao & Bilen, 2021) data augmentation during the evaluation model training process. Other parameters are set in accordance with those used in BACON (Zhou et al., 2024b), ensuring consistency across experiments. The overall experimental setup follows the guidelines outlined in BEARD (Zhou et al., 2024a). All experiments, including synthetic dataset generation and model training, are conducted on NVIDIA RTX 3090 GPU clusters.

### C.2. Adversarial Robustness Evaluation

**White-box Robustness.** The robustness curves are shown in Figure 5. We evaluate the adversarial robustness of models trained on distilled datasets with different IPC settings using PGD attacks under both targeted and untargeted settings, across various perturbation budgets. As the perturbation budget increases, the adversarial robustness of all models decreases, with the rate of decline serving as an indicator of robustness. Notably, the model trained with ROME exhibits the slowest decrease in accuracy, demonstrating superior robustness compared to other methods. Furthermore, the robustness curves under targeted attacks decline more gradually than those under untargeted attacks, indicating that ROME is particularly more robust against targeted PGD attacks. This enhanced robustness aligns with the incorporation of targeted PGD as a robustness prior during training, which likely contributes to its improved defense against such attacks.

**Black-box Robustness.** Table 5 reports the I-RR of ROME and baseline methods under transfer-based black-box attacks on CIFAR-10, evaluated at IPC levels 1, 10, 50, and the aggregated metric M. While robustness generally decreases with increasing IPC for all methods and is higher under targeted than untargeted attacks, ROME consistently achieves the highest robustness with the smallest variation across IPC settings and attack types. This demonstrates ROME's superior and stable adversarial robustness across different dataset distillation scales and attack scenarios.

### C.3. Additional Results on Adversarially Distilled Dataset Training Efficiency

**CREI Comparison and Robustness Analysis.** We compare ROME against adversarially distilled datasets using the BEARD benchmark and I-CREI as a unified robustness metric. As summarized in Table 6, ROME delivers strong adversarial robustness under targeted attacks across different dataset compression levels (IPC-1, 10, 50) and their aggregation (denoted

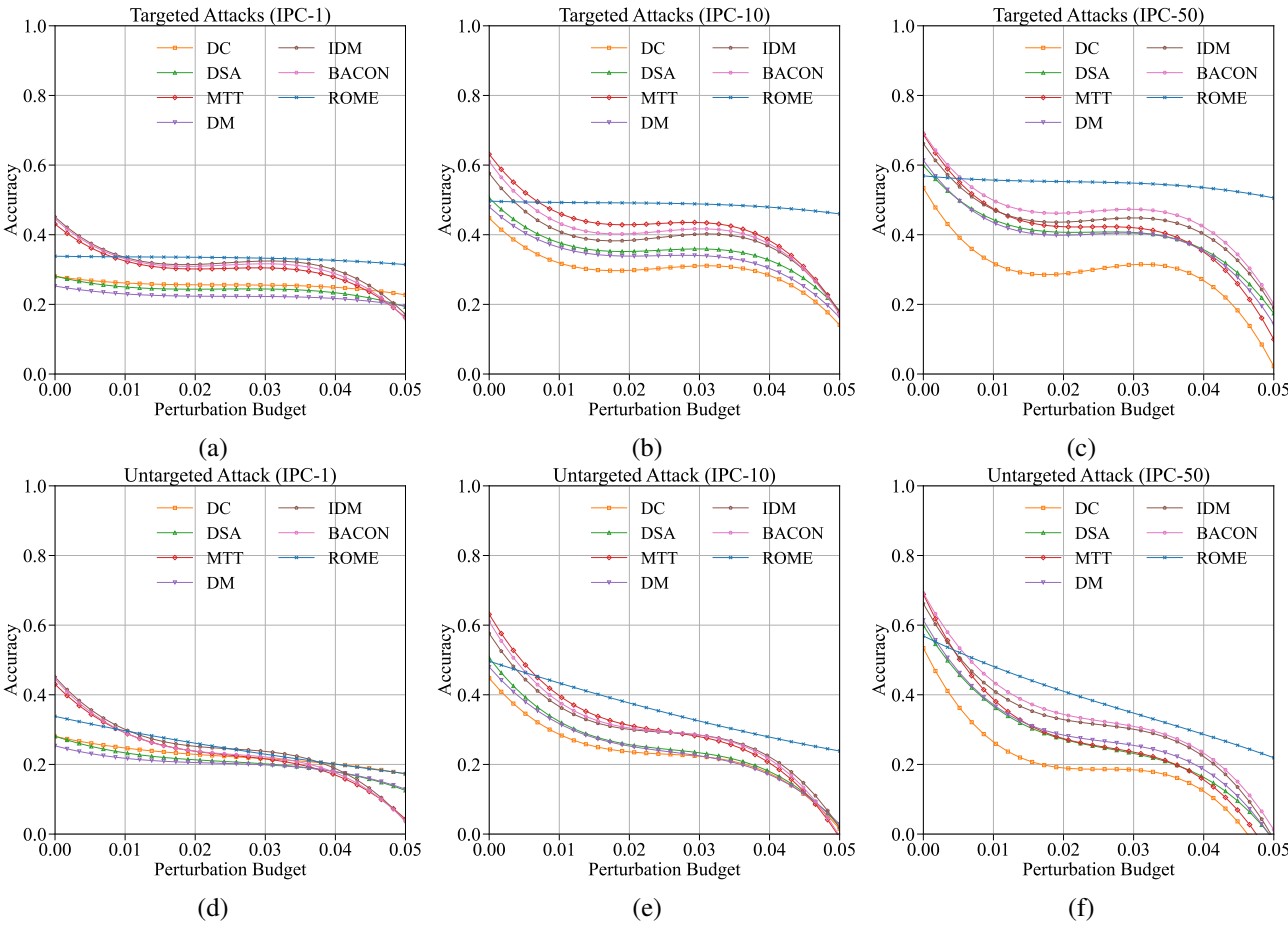

*Figure 5.* Robustness curves of models trained on distilled CIFAR-10 datasets with varying IPC settings under white-box attacks. Subfigures (a), (b), and (c) present the results for targeted attacks with IPC-1, IPC-10, and IPC-50, respectively, while (d), (e), and (f) show the corresponding results for untargeted attacks with the same IPC settings.

*Table 5.* Comparison of I-RR across IPC settings 1, 10, 50, and M (a unified metric aggregating these settings) for various DD methods under black-box attacks on CIFAR-10. Best results are highlighted in **bold**.

| Method | Targeted Attack | | | | Untargeted Attack | | | |
|---|---|---|---|---|---|---|---|---|
| | IPC-1 | IPC-10 | IPC-50 | IPC-M | IPC-1 | IPC-10 | IPC-50 | IPC-M |
| DC | 99.52% | 92.54% | 85.84% | 90.67% | 99.28% | 90.03% | 83.97% | 89.31% |
| DSA | 98.65% | 93.81% | 94.09% | 90.92% | 98.30% | 91.11% | 92.31% | 89.13% |
| MTT | 95.31% | 93.40% | 91.40% | 89.80% | 94.05% | 90.63% | 89.02% | 87.52% |
| DM | 99.15% | 94.53% | 92.22% | 90.08% | 99.12% | 91.02% | 90.36% | 88.45% |
| IDM | 93.86% | 92.12% | 92.17% | 89.91% | 91.88% | 88.99% | 89.22% | 86.97% |
| BACON | 93.80% | 91.90% | 92.46% | 89.24% | 91.18% | 88.46% | 89.25% | 85.66% |
| ROME | **99.98%** | **99.98%** | **99.90%** | **99.44%** | **99.60%** | **98.94%** | **98.44%** | **97.34%** |

*Table 6.* Comparison of adversarial robustness between ROME and adversarially distilled datasets using DD methods under CIFAR-10 IPC-1, IPC-10, and IPC-50, evaluated using I-CREI against both targeted and untargeted attacks. Best results are highlighted in **bold**.

| Mrthod | Targeted Attack | | | | Untargeted Attack | | | |
|---|---|---|---|---|---|---|---|---|
| | IPC-1 | IPC-10 | IPC-50 | IPC-M | IPC-1 | IPC-10 | IPC-50 | IPC-M |
| DC | **64.51%** | 64.27% | 63.96% | 63.43% | **56.44%** | 51.52% | 49.60% | **48.08%** |
| DSA | 64.15% | 64.09% | 64.47% | **63.46%** | 53.54% | 50.09% | 50.73% | 47.19% |
| MTT | 63.05% | 63.90% | 63.98% | 62.44% | 47.50% | 49.58% | 49.52% | 44.39% |
| DM | 64.34% | 64.00% | **64.54%** | 63.21% | 53.52% | 50.62% | 51.89% | 46.58% |
| IDM | 64.47% | 64.09% | 64.29% | 63.11% | 51.19% | **51.63%** | **51.99%** | 47.96% |
| BACON | 63.84% | 64.03% | 64.22% | 62.68% | 47.37% | 50.02% | 51.32% | 44.66% |
| ROME | 64.46% | **64.32%** | 63.66% | 63.32% | 51.24% | 48.29% | 43.67% | 43.62% |

*Table 7.* Comparison of training time for ROME and adversarially distilled datasets using DD methods on CIFAR-10 IPC-50. Best results are highlighted in **bold**. The training time is measured in hours, with total time including both standard training and adversarial retraining.

| Method | Standard Training | Adversarial Training | Total |
|---|---|---|---|
| DC | 0.425 | 0.663 | 1.088 |
| DSA | 0.437 | 0.666 | 1.103 |
| MTT | 0.444 | 0.644 | 1.088 |
| DM | 0.452 | 0.658 | 1.109 |
| IDM | 0.414 | 0.641 | 1.055 |
| BACON | 0.442 | 0.659 | 1.101 |
| ROME | **0.418** | **0.000** | **0.418** |

as IPC-M), achieving comparable or better results than prior methods such as DC, DSA, MTT, DM, IDM, and BACON. Notably, unlike these baselines which require adversarial retraining to reach competitive robustness, ROME attains its performance without any additional adversarial fine-tuning.

Under untargeted attacks, ROME exhibits a decline in robustness at higher IPCs (e.g., 43.67% at IPC-50), likely due to its reliance on robust priors generated via targeted PGD during distillation, which biases the model toward robustness against targeted threats. As further discussed in Section 4.2, this design can lead to an **Over-Robustness Phenomenon**, where improvements in targeted robustness do not fully generalize to untargeted settings. Consequently, ROME demonstrates weaker robustness under untargeted attacks compared to adversarially trained DD methods. Despite this limitation, ROME strikes a favorable balance between robustness and training efficiency, making it a lightweight and practical distillation approach. Future work will aim to design distillation objectives that improve robustness across various threat models.

**Training Time Comparison and Efficiency Analysis.** Table 7 compares the training times for neural network models trained on distilled datasets generated by ROME and several other dataset distillation methods, including DC, DSA, MTT, DM, IDM, and BACON, under CIFAR-10 IPC-50. ROME requires 0.418 hours for model training on the distilled dataset, which is slightly higher than IDM at 0.414 hours. However, ROME does not involve adversarial retraining. In contrast, methods such as DC, DSA, MTT, DM, and IDM necessitate additional adversarial retraining to achieve comparable robustness. For example, DM requires a total of 1.109 hours, with 0.658 hours spent on adversarial retraining, more than doubling the training time compared to standard training.

The primary computational cost for ROME stems from distilling adversarially robust features, which forces the model to learn from a broader set of adversarial examples during the training process. While this adds complexity to the distillation process, ROME continues to perform excellently during standard training. Unlike methods such as DC, DSA, MTT, DM, and IDM, which require additional adversarial retraining to achieve comparable robustness, ROME eliminates this retraining step, significantly reducing computational overhead. This makes ROME a more efficient approach, as it achieves strong adversarial robustness without the time-consuming retraining phase.

*Table 8.* Ablation study of Robust Pretrained Model (RPM) and Adversarial Perturbation (AP) on CIFAR-10 with IPC-50, evaluated under targeted and untargeted attacks using I-RR, AE, and I-CREI. Best results are highlighted in **bold**.

| Baseline | RPM ($f(\cdot)$) | AP ($\hat{x}$) | Targeted Attack | | | Untargeted Attack | | |
|---|---|---|---|---|---|---|---|---|
| | | | I-RR | AE | I-CREI | I-RR | AE | I-CREI |
| ✓ | | | 81.86% | 28.66% | 55.26% | 32.45% | 26.13% | 29.29% |
| ✓ | ✓ | | 84.50% | 28.56% | 56.53% | 34.89% | 26.00% | 30.45% |
| ✓ | | ✓ | 94.66% | 28.68% | 61.67% | 47.64% | 25.92% | 36.78% |
| ✓ | ✓ | ✓ | **97.73%** | **28.73%** | **63.23%** | **51.73%** | **26.16%** | **38.95%** |

## C.4. Ablation Study

More details are provided in Table 8, which also includes the calculation of the Attack Efficiency Ratio (AE). The ablation studies confirm that ROME incorporates the best-performing results.

## C.5. Visualization

Figure 6 shows the distilled datasets generated by ROME under varying robust prior configurations, highlighting their impact on the synthetic data distribution. Figures 7, 8, and 9 illustrate the distilled datasets generated by ROME, BACON, and IDM on CIFAR-10 and CIFAR-100 with IPC settings of 50, 10, and 1, respectively.

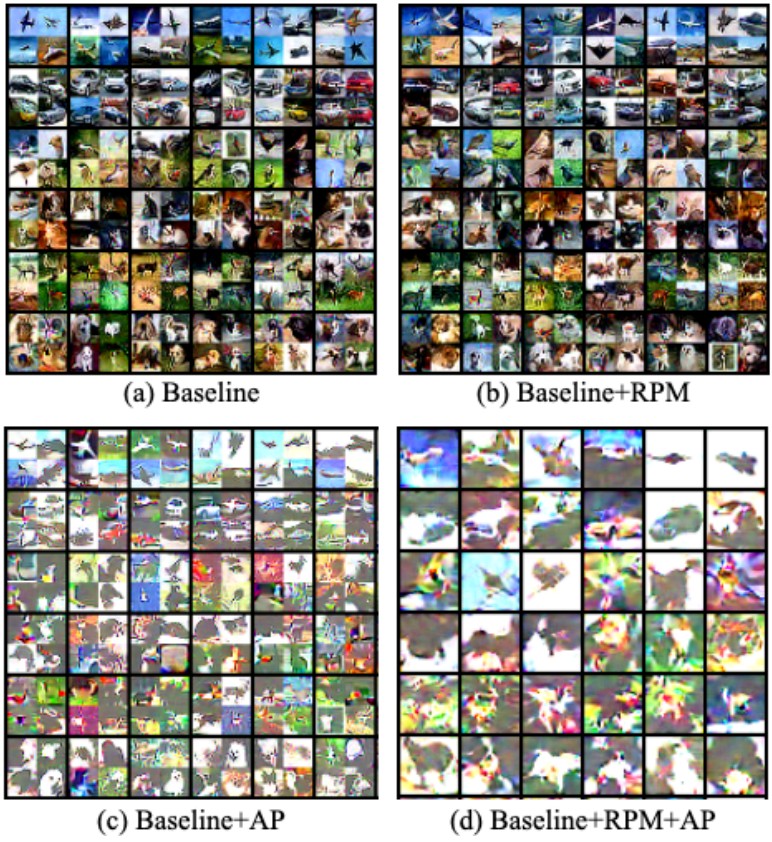

*Figure 6.* Visualization of distilled datasets generated by ROME under different robust prior configurations, showcasing the impact of varying settings on the synthetic data distribution.

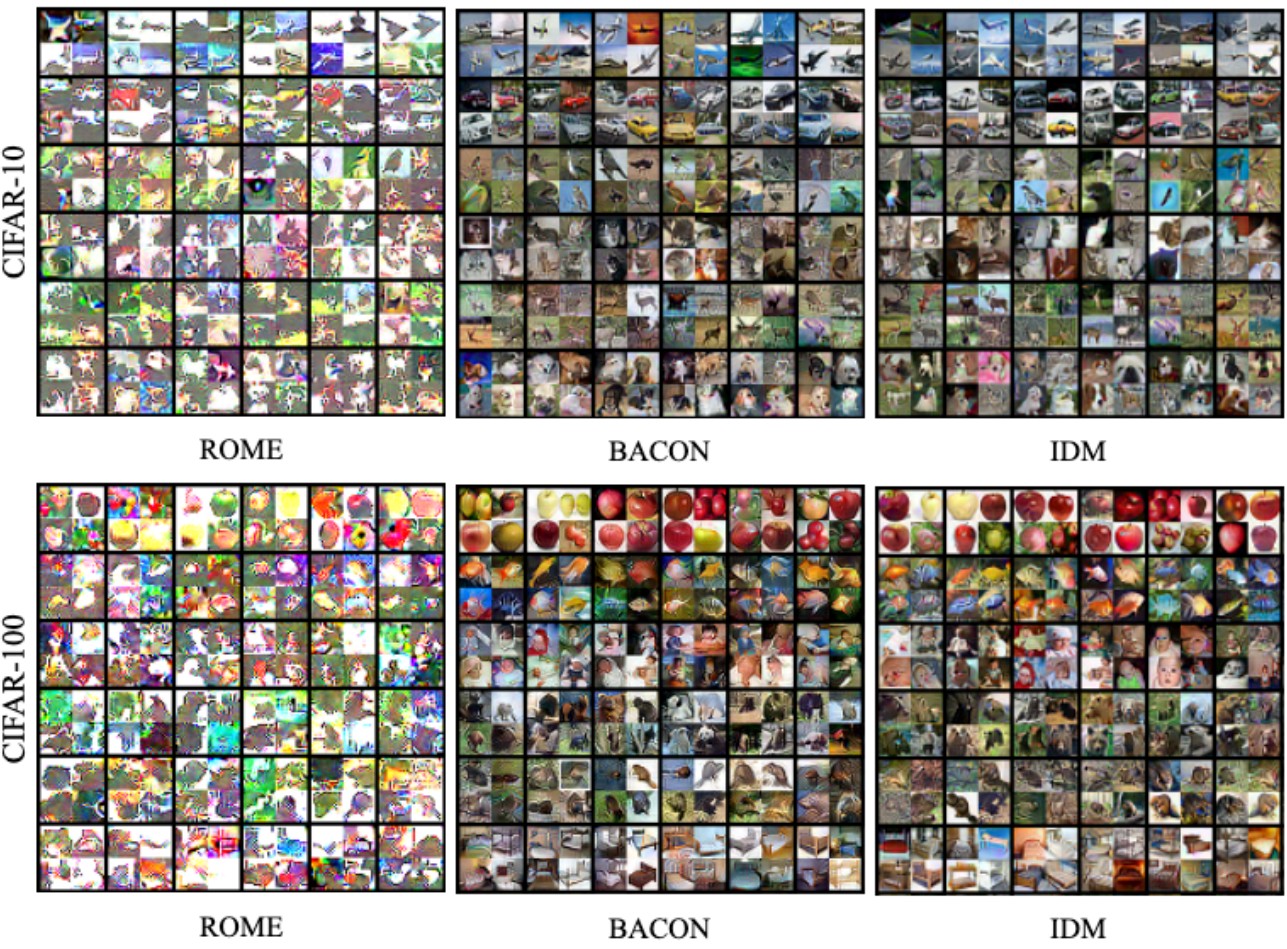

*Figure 7.* Visualizations of distilled datasets generated by diverse DD methods with IPC-50 settings on the CIFAR-10 and CIFAR-100.

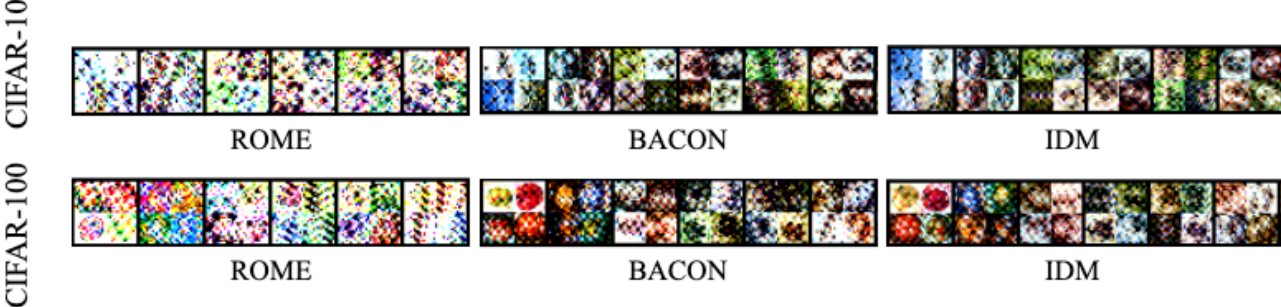

*Figure 8.* Visualizations of distilled datasets generated by diverse DD methods with IPC-10 settings on the CIFAR-10 and CIFAR-100.

*Figure 9.* Visualizations of distilled datasets generated by diverse DD methods with IPC-1 settings on the CIFAR-10 and CIFAR-100.

