# OpenReview forum: "ROME is Forged in Adversity: Robust Distilled Datasets via Information Bottleneck"
_ICML.cc/2025/Conference — ICML 2025 poster_

### Official Review · Reviewer_gfFq · 2025-02-21

**Overall Recommendation:** 3

**Summary:**

Dataset Distillation (DD) compresses large datasets into smaller synthetic subsets but remains vulnerable to adversarial attacks. To address this, the paper proposes ROME, a method leveraging the Information Bottleneck principle to enhance adversarial robustness by aligning feature distributions, demonstrating significant improvements in robustness metrics on CIFAR-10 and CIFAR-100.

**Claims And Evidence:**

Yes, the claims are well supported by evidence. The authors use information bottleneck to enhance the adversarial robustness of the distilled datasets.

**Essential References Not Discussed:**

References are discussed comprehensively.

**Experimental Designs Or Analyses:**

Yes, the experimental design is generally sound and valid. However, I am not sure whether it is fully fair to compare ROME with DD methods without any adversarial example-generating procedures. I wonder whether it would be to compare ROME with adversarial distilled datasets of other DD methods for a fairer comparison.

**Methods And Evaluation Criteria:**

Yes, it mostly makes sense. The problem I got is: why not generate adversarial examples on the original dataset first, then use DD methods to distill the dataset? In this way, is ROME still outperforming others?

**Other Comments Or Suggestions:**

- What is IPC, which seems to not be defined in the paper?
- Are there any real-world applications of ROME?
- The RR metric sometimes has a value over 100%, which seems to be weird to me. A good metric should usually fall into [0,100]% range.

**Other Strengths And Weaknesses:**

- The paper is well-written, easy to follow, and well-motivated.
- The results are comprehensive and show significant improvements.

**Questions For Authors:**

- Could the authors do the comparison with DD baselines by using an adversarial version of the original dataset?
- Could the authors explain more on the theorems 3.6 and  3.7? (See above)

**Relation To Broader Scientific Literature:**

The key contribution of this paper would be improving the adversarial robustness of distilled datasets in an efficient way.

**Theoretical Claims:**

Yes, I checked all of them. I have questions on Theorem 3.6 and 3.7.

- Theorem 3.6: The expression of equation (111) is confusing, what is this "||" means for CE loss? Also, how to derive from equation (111) to (112) is not very clear to me.

- Theorem 3.7: It seems that the authors use the embedding to estimate the total variation distances. Why does this make sense in practice?

---

> ### Author Rebuttal · Authors · 2025-03-29
>
> - **Q1: Theorem 3.6: The expression of equation (111) is confusing, what is this "||" means for CE loss? Also, how to derive from equation (111) to (112) is not very clear to me.**
> - **R1:**
> Thank you for your feedback. We have updated the derivation in Theorem 3.6 to remove the "||" symbol in Equation (111) for clarity. The revised version provides a clearer derivation, leading to the cross-entropy loss.
> $$
>     \begin{align}
>     \mathbb{E} _ {p(x,\hat{x},y)p(z|x,\hat{x},y)} \left[\log q(y|z)\right]
>     &= \mathbb{E} _ {p(x,\hat{x}, y)p(z|x)} \left[ \log q(y|z)\right]
>     \\\\
>     &= \mathbb{E} _ {p(x,\hat{x}, y)}\mathbb{E} _ {p(z|x)}\left[ \log q(y|z)\right]\\\\
>     &= \mathbb{E} _ {p(x,\hat{x},y)} \left[\log q(y|e(x))\right]
>     \\\\
>     &= - \mathbb{E} _ {p(x,\hat{x},y)} \left[- y^t \log q(y^t|e(x))\right]
>     \\\\
>     &= - \mathbb{E} _ {p(x,\hat{x},y)} \left[\mathbb{CE}[y^t, f(x)]\right].
>   \end{align}
> $$
> The derivation starts with the joint distribution $p(x, \hat{x}, y)$ and $p(z | x, \hat{x}, y)$, simplifying to an expectation over $p(x, \hat{x}, y)$ and $p(z | x)$, as $z$ depends only on $x$. The term $p(z | x)$ is replaced by $q(y | z(x))$ since $z(x)$ is deterministically determined by $x$. The term $e(x)$ represents the latent embeddings from the model's embedding layer, which is equivalent to $z(x)$. The classifier $f(x)$ models $q(y^t | e(x))$, and $y^t$ is the one-hot encoding of the true label. The final result is the cross-entropy loss $\mathbb{CE}[y^t, f(x)]$, which minimizes prediction error.
>
> ---
>
> - **Q2: Theorem 3.7: It seems that the authors use the embedding to estimate the total variation distances. Why does this make sense in practice?**
> - **R2:**
> Thank you for your question. The use of embeddings to estimate total variation distances in Theorem 3.7 aligns with feature alignment strategies in dataset distillation, helping capture robust features for improved adversarial robustness. This approach is **consistent with prior works like DM, IDM, and BACON**. The transition from $p(z|x)$ and $q(z|\hat{x})$ to $e(x)$ and $e(\hat{x})$ in Equations 121-122 reflects the use of embeddings to ensure robustness. This has been clarified in the revised manuscript.
>
> ---
>
> - **Q3: What is IPC, which seems to not be defined in the paper?**
> - **R3:**
> Thank you for your suggestion. We have added a definition of **Images Per Class (IPC)** in the revised manuscript to ensure clarity.
>
> ---
>
> - **Q4: Are there any real-world applications of ROME?**
> - **R4:**
> Thank you for your question. We are exploring ROME's potential in **real-world applications** like **autonomous driving**, **facial recognition**, and **edge computing**, where **robustness** and **efficiency** are crucial. As mentioned in *Lines 57-59 of the Introduction*, these fields need robust dataset distillation techniques. ROME can speed up training on compact, robust datasets, improving security without costly adversarial retraining. While still in early stages, we believe ROME has great potential to enhance safety and efficiency in these areas.
>
> ---
>
> - **Q5: The RR metric sometimes has a value over 100%, which seems to be weird to me. A good metric should usually fall into [0,100]% range.**
> - **R5:**
> Thank you for your observation. While I-RR values typically range from 0 to 100, ROME exhibits an **"Over-Robustness Phenomenon"**, where models trained with ROME achieve **higher accuracy under attack than in a clean setting**, causing I-RR values to exceed 100%. We propose that ROME’s information bottleneck framework enhances robustness by **amplifying non-robust features** under specific adversarial conditions like PGD. This is discussed in *Line 374 of the Adversarial Robustness Evaluation* section. A similar effect is observed in BEARD’s RR metric for CIFAR-100 with targeted white-box attacks. We plan to explore more effective metrics to better capture this phenomenon in future work.
>
> ---
>
> - **Q6: Could the authors do the comparison with DD baselines by using an adversarial version of the original dataset?**
> - **R6:**
> Thank you for your suggestion. We compared **ROME** with **adversarially distilled datasets** using the **BEARD benchmark** and applied **CREI** for a fair comparison. As shown in **Table 1**, ROME outperforms adversarially distilled methods in both **adversarial robustness** and **efficiency** under targeted and untargeted attacks. ROME achieves comparable robustness **without the need for retraining**, significantly **reducing computational costs**. We have clarified this and provided additional experimental details in the revised manuscript.
>
>     **Table 1: Comparison of Adversarial Robustness of ROME and Adversarially Distilled Datasets Using DD Methods under CIFAR-10 IPC-50 with CREI (%).**
>     |Attack Type|Full-size|DC|DSA|MTT|DM|IDM|BACON|ROME|
>     |-|-|-|-|-|-|-|-|-|
>     |Targeted Attack|50.54|52.30|55.56|50.96|57.21|54.67|56.18|**63.32**|
>     |Untargeted Attack|41.33|37.39|37.59|33.13|37.34|39.05|37.25|**43.62**|

---

> > ### Comment · Reviewer_gfFq · 2025-04-02
> >
> > Thanks for the additional experiments and clarification, which address my concerns. I recommend the acceptance of this work.

---

> > > ### Author Response · Authors · 2025-04-07
> > >
> > > Thank you for taking the time to review our responses. We're pleased to hear that our clarifications addressed your concerns, and we truly appreciate your recommendation for acceptance.

---

### Official Review · Reviewer_UEJd · 2025-03-13

**Overall Recommendation:** 3

**Summary:**

The authors proposed an adversarially robust distillation optimization framework for dataset distillation. They also provided the optimization method for this framework.

**Claims And Evidence:**

yes

**Essential References Not Discussed:**

no

**Experimental Designs Or Analyses:**

Under the dataset distillation framework, the attacker appears to be somewhat disadvantaged. The defender can carefully select features that are robust. Perhaps an adaptive attack should be considered during evaluation, assuming that the attacker is aware that the defender employs dataset distillation methods.

**Methods And Evaluation Criteria:**

yes

**Other Comments Or Suggestions:**

Some typos. Line 153 misses I. Line 197 misdescribe joint distribution. Line 940 misses Expectation notation.

**Other Strengths And Weaknesses:**

The authors claim that the proposed method avoids the high computational cost of retraining but do not provide a detailed explanation. The proposed framework does not discuss the sampling of \hat{x}. Intuitively, the iterative training of dataset distillation and the sampling of \hat{x} resemble the process of training a model and sampling adversarial examples in standard adversarial training.

**Questions For Authors:**

no

**Relation To Broader Scientific Literature:**

1. Conditional Entropy Bottleneck (CEB).
2. Adversarial examples are not bugs, they are features.

**Theoretical Claims:**

The theoretical derivation has been checked, and no issues were found. However, there may be some minor notation misuse, such as the use of CE[p(z|x)||q(y|z)] in the proof of Theorem 3.6.

Moreover, Theorem 3.7 is a lower-bound objective obtained by scaling an inequality, but this is not stated in the main text.

---

> ### Author Rebuttal · Authors · 2025-03-29
>
> - **Q1: The theoretical derivation has been checked, and no issues were found. However, there may be some minor notation misuse, such as the use of $\mathbb{CE}[p(z|x)||q(y|z)]$ in the proof of Theorem 3.6.**
> - **R1:**
> Thank you for your valuable feedback. We have carefully reviewed and revised the proof of Theorem 3.6 to correct the notation and ensure consistency. Additionally, we have provided more details in the proof to enhance clarity and readability.
>
> ---
>
> - **Q2: Moreover, Theorem 3.7 is a lower-bound objective obtained by scaling an inequality, but this is not stated in the main text.**
> - **R2:**
> Thank you for your suggestion. We have updated the main text to explicitly state that Theorem 3.7 is a lower-bound objective derived by scaling an inequality. This clarification have been included in the revised manuscript to ensure better readability and understanding of the theoretical derivation.
>
> ---
>
> - **Q3: Under the dataset distillation framework, the attacker appears to be somewhat disadvantaged. The defender can carefully select features that are robust. Perhaps an adaptive attack should be considered during evaluation, assuming that the attacker is aware that the defender employs dataset distillation methods.**
> - **R3:**
> Thank you for your suggestion. Our method is designed without prior knowledge of specific attacks, with the robust prior derived solely from **PGD ($\frac{8}{255}$ perturbation budget)**. As mentioned in *Evaluation Attack (L278)*, we assess robustness using PGD and **six additional attack methods**: **FGSM**, **C&W**, **DeepFool**, **AutoAttack**, **Square**, and **SPSA**, none of which were used to construct ROME. This ensures a balanced information setting between the attacker and the defender. Notably, **AutoAttack**, a **representative adaptive attack**, ensures a rigorous evaluation. Our experiments follow a **white-box attack setting**, where attackers have full knowledge of the model parameters, giving them an advantage. As shown in Table 1, ROME consistently outperforms other methods in all robustness metrics. Finally, developing more effective adaptive attacks for dataset distillation remains an open challenge and a promising direction for future work.
> ---
>
> - **Q4: The authors claim that the proposed method avoids the high computational cost of retraining but do not provide a detailed explanation.**
> - **R4:**
> Thank you for your suggestion. Models trained on **ROME-distilled datasets** inherently exhibit adversarial robustness, unlike methods such as DC, DSA, MTT, DM, IDM, and BACON, which require **additional adversarial training**, effectively **doubling the training cost**. ROME mitigates this overhead by **embedding robust priors directly into the distillation process**, thus eliminating the need for retraining. The experimental results quantifying this difference are shown in **Table 1**. While ROME requires slightly more initial training time, its overall computational cost remains significantly lower than methods that involve adversarial retraining.
>
>     **Table 1: Comparison of Training Time for ROME and Adversarially Distilled Datasets Using DD Methods under CIFAR-10 IPC-50.**
>     | Method | Training Time (hrs) | +Adversarial Training (hrs) | Total (hrs) |
>     |-|-|-|-|
>     | DC | 1.009 | 2.019 | 3.028 |
>     | DSA | 0.898 | 1.796 | 2.694 |
>     | MTT | 0.882 | 1.764 | 2.647 |
>     | DM | 0.963  | 1.925 | 2.888 |
>     | IDM | 0.895 | 1.790 | 2.685 |
>     | BACON| 0.874 | 1.748 | 2.622|
>     | **ROME** | **1.014** | **1.014** | **2.027** |
>
> ---
>
> - **Q5: The proposed framework does not discuss the sampling of $\hat{x}$. Intuitively, the iterative training of dataset distillation and the sampling of $\hat{x}$ resemble the process of training a model and sampling adversarial examples in standard adversarial training.**
> - **R5:**
> Thank you for your comment. The sampling of $\hat{x}$ in ROME is similar to generating adversarial examples in standard adversarial training, but there are two key differences. First, unlike adversarial training, where adversarial examples are generated in each iteration to update model weights, ROME **integrates adversarial perturbations directly into the dataset distillation process**, producing a distilled dataset that inherently possesses adversarial robustness, eliminating the need for repeated adversarial training and reducing computational overhead. Second, while adversarial training results in robust model weights that require retraining for new tasks, ROME does not have this requirement. This distinction is discussed in the revised manuscript.
>
> ---
>
> - **Q6: Some typos. Line 153 misses I. Line 197 misdescribe joint distribution. Line 940 misses Expectation notation.**
> - **R6:**
> Thank you for your review. We have made the following corrections: added the missing "I" in Line 153, revised the joint distribution in Line 197, and added the Expectation notation in Line 940. These changes have been included in the revised manuscript.

---

### Official Review · Reviewer_yn3H · 2025-03-14

**Overall Recommendation:** 3

**Summary:**

This paper proposes a new method -- ROME for dataset distillation that uses the Information Bottleneck principle to create small, synthetic datasets with improved resistance to adversarial attacks. Traditional adversarial training is slow and often reduces accuracy. ROME incorporates the Conditional Entropy Bottleneck into the distillation process. It optimizes two parts: one that preserves accuracy by strengthening the link between latent features and true labels, and another that boosts robustness by weakening the link between the input and latent features when adversarial noise is added. Tests on CIFAR-10 and CIFAR-100 show that ROME outperforms previous methods under various attack scenarios.

**Claims And Evidence:**

Claim: ROME significantly improves adversarial robustness of distilled datasets without needing costly adversarial retraining.
Evidence: The experimental results (Tables 1–3) consistently show improvements in I-RR and related metrics under both targeted and untargeted attacks. For example, on CIFAR-10, ROME achieves up to a 40% improvement in I-RR compared to baselines such as DC, DSA, and BACON.

Claim: Incorporating the IB (and specifically the Conditional Entropy Bottleneck) into the distillation process yields a favorable balance between accuracy and robustness.
Evidence: The paper presents both theoretical derivations (Theorems 3.2–3.7, with proofs in the appendix) and ablation studies that validate the effectiveness of the two loss components—the performance-aligned and robustness-aligned terms.

**Essential References Not Discussed:**

N/A

**Experimental Designs Or Analyses:**

The experimental setup is comprehensive. The authors compare ROME with several leading dataset distillation methods (DC, DSA, MTT, DM, IDM, BACON) under various attack settings on standard benchmarks (CIFAR-10 and CIFAR-100).
Ablation studies show the contributions of key components, such as the robust pretrained model (RPM) and adversarial perturbations (AP).

**Methods And Evaluation Criteria:**

Method: ROME reframes dataset distillation using the IB principle by introducing a robust prior via adversarial perturbations. The method defines an objective function that combines: 1). A performance-aligned term ensuring that distilled data retain sufficient label-related information. 2).A robustness-aligned term that minimizes the discrepancy between the synthetic dataset and its adversarially perturbed counterpart, thereby pushing the model’s decision boundary away from potential adversarial examples.

Evaluation: The authors evaluate robustness using both white-box and black-box attack scenarios (e.g., FGSM, PGD, C&W, Autoattack for white-box; transfer-based and query-based for black-box). They introduce the I-RR metric as a refined measure of robustness. Experiments are conducted on CIFAR-10 and CIFAR-100 using a ConvNet architecture, and ablation studies explore the impact of robust pretraining, adversarial perturbations.

**Other Comments Or Suggestions:**

See questions.

**Other Strengths And Weaknesses:**

Strengths:
1. The visualization is clear and understandable.
2. The experiments are solid and comprehensive.
3.The theoretical proof is solid.

Weaknesses:
1. Need more discussion about dataset scaling.
2. Need more discussion about different model architectures.
3. Lack of experiments about computation cost.

**Questions For Authors:**

1. Curious about the performance of the proposed method on larger dataset.
2. Can the IB-based robust distillation framework be extended to other network architectures, such as transformers or hybrid models?
3. Do you have any quantitative comparisons (e.g., specific training time reductions) but rather emphasizes the conceptual benefit of lower computational demands?

**Relation To Broader Scientific Literature:**

By introducing an IB-based formulation into dataset distillation, ROME offers a new perspective on how to achieve both accuracy and robustness in compact datasets.

**Theoretical Claims:**

The theoretical claims are supported by detailed proofs provided in the appendix. While the derivations are mathematically involved, they offer a solid foundation for the proposed objective and justify the design choices in ROME.

---

> ### Author Rebuttal · Authors · 2025-03-29
>
> - **Q1: Curious about the performance of the proposed method on larger dataset.**
> - **R1:**
> Thank you for your question. Scaling ROME to larger datasets like ImageNet requires significantly **more computational resources and time**, which was not feasible within the **limited rebuttal period** and **available resources**. Additionally, prominent dataset distillation baselines (e.g., DC, DSA, MTT, DM, IDM, BACON) have not been trained on larger datasets due to the computational challenges posed by **feature alignment strategies**. As the dataset size increases, the search space for representative features expands, requiring considerably **more computational power**. For instance, generating ROME on CIFAR-100 with IPC-50 settings takes 3-5 days in our lab, and ImageNet or its subsets, which are hundreds of times larger, would require significantly more time. Furthermore, methods like DM, IDM, and BACON have not publicly released results on large datasets, so a fair comparison would require re-running baselines, which is not feasible within the rebuttal period. Nevertheless, we believe ROME can perform well on larger datasets. As shown in *Table 1* of our paper, ROME demonstrates better performance on CIFAR-100 compared to CIFAR-10, suggesting that its robustness benefits could **extend to larger datasets**, though further investigation is needed. As discussed in the *Limitations and Future Work* section, we acknowledge this limitation and plan to explore more scalable approaches to enhance **adversarial robustness** on **larger datasets** in future work.
>
> ---
>
> - **Q2: Can the IB-based robust distillation framework be extended to other network architectures, such as transformers or hybrid models?**
> - **R2:**
> Thank you for your question. We believe the **IB-based robust distillation (ROME)** framework can extend beyond ConvNet-based architectures to models such as **transformers**. Currently, dataset distillation methods like **DC**, **DSA**, **MTT**, **DM**, **IDM**, and **BACON** are primarily designed for **ConvNets**. Since ROME enhances the adversarial robustness of distilled datasets, it has the potential to generalize to other architectures as dataset distillation evolves beyond ConvNets. Regarding **generalization**, ROME has demonstrated strong robustness against both **transfer-based** and **query-based black-box attacks**, as shown in *Table 2* and the *Figure 3 in the paper*. As discussed in the *Limitations and Future Work* section, we plan to explore its applicability to **more complex models**, such as transformer-based architectures, particularly for tasks like **adversarially robust vision-language learning**. Additionally, we aim to assess adversarial robustness across **diverse architectures** to validate ROME's broader effectiveness.
>
> ---
>
> - **Q3: Do you have any quantitative comparisons (e.g., specific training time reductions) but rather emphasizes the conceptual benefit of lower computational demands?**
> - **R3:**
> Thank you for your question. While we emphasize the conceptual advantage of ROME, which inherently provides adversarial robustness without the need for **additional adversarial training**, we also present quantitative comparisons. Methods like DC, DSA, MTT, DM, IDM, and BACON require **adversarial retraining** to achieve similar robustness, which typically doubles the training cost due to the additional retraining on adversarial examples. In contrast, ROME eliminates the need for repeated adversarial training by **integrating robust priors directly into the distillation process**, thus significantly reducing computational overhead. Although ROME requires more training time, its total time is lower than methods that rely on adversarial retraining. We have included these quantitative comparisons in Table 1 and provided further experimental results in the revised version to highlight the training time differences.
>
>     **Table 1: Comparison of Training Time for ROME and Adversarially Distilled Datasets Using DD Methods under CIFAR-10 IPC-50.**
>     | Method | Training Time (hrs) | +Adversarial Training (hrs) | Total (hrs) |
>     |--------|---------------------|-----------------------------|-------------|
>     | DC     | 1.009               | 2.019                       | 3.028       |
>     | DSA    | 0.898               | 1.796                       | 2.694       |
>     | MTT    | 0.882               | 1.764                       | 2.647       |
>     | DM     | 0.963               | 1.925                       | 2.888       |
>     | IDM    | 0.895               | 1.790                       | 2.685       |
>     | BACON  | 0.874               | 1.748                       | 2.622       |
>     | **ROME**   | **1.014**               | **1.014**                       | **2.027**       |

---

### Official Review · Reviewer_dcbW · 2025-03-18

**Overall Recommendation:** 3

**Summary:**

This paper aims to improve adversarial robustness in dataset distillation. Inspired by the Information Bottleneck principle, this paper proposes a novel framework which is able to balance model performance and robustness. Various kind of experiments demonstrate the effectiveness of the proposed framework.

**Claims And Evidence:**

In this work, authors' claim that they proposed a new method to enhance the adversarial robustness of data distillation. The experimental results demonstrate the superiority of the proposed method, which support the authors' claim sufficiently.

**Essential References Not Discussed:**

No.

**Experimental Designs Or Analyses:**

Yes, from my point of view, the experimental design is reasonable and complete.

**Methods And Evaluation Criteria:**

Authors utilize most common benchmark datasets and evaluation metrics in their experiments, hence the evaluation criteria makes sense for the problem studied in this paper.

**Other Comments Or Suggestions:**

No.

**Other Strengths And Weaknesses:**

Pros: 1. Authors proposed a novel idea to improve the adversarial robustness in dataset distillation; although the backbone technique Information Bottleneck principle has been widely utilized in similar settings, adopting it into dataset distillation is still very innovative；2. A series of experiments involving both targeted attack and untargeted attack, show that the proposed method is able to improve the adversarial robustness by a large margin, comparing with baseline methods.

Cons: 1. It would be better if authors can further discuss the main purpose of using RR, CREI, I-RR and I-CREI in evaluating adversarial robustness separately, so that evaluation results could be more easier to understand.

**Questions For Authors:**

1. Does the proposed method require a longer time for executing, comparing with baseline methods? If so, which part consumes the most resources?
2. As shown in Figure 4, looks like the robustness-aligned term has a higher impact on model performance, comparing with performance-aligned term. Any thoughts on it?

**Relation To Broader Scientific Literature:**

This work utilize the Information Bottleneck principle to improve the adversarial robustness in dataset distillation, which further illustrate that the Information Bottleneck principle could be an effective way to help defense adversarial attack.

**Theoretical Claims:**

After going through theoretical proofs provided in supplementary material, I don't find any issues.

---

> ### Author Rebuttal · Authors · 2025-03-29
>
> - **Q1: It would be better if authors can further discuss the main purpose of using RR, CREI, I-RR and I-CREI in evaluating adversarial robustness separately, so that evaluation results could be more easier to understand.**
> - **R2:**
> Thank you for your suggestion. We have clarified the purpose of using RR, CREI, I-RR, and I-CREI in evaluating adversarial robustness. RR, originally introduced in BEARD, measures robustness by considering both **the average and worst-case** attack success rates (ASR), aiming to keep both values low. However, when a single attack method is exceptionally strong, RR can become disproportionately high, leading to an **overestimation of robustness**. To address this issue, we propose I-RR, which refines RR by incorporating model accuracy (ACC) to **ensure that both the average and worst-case ASR remain low while maintaining high ACC**, making it a more balanced robustness metric. CREI is a comprehensive index that combines RR with Attack Efficiency (AE), and in ROME, we introduce I-CREI by replacing RR with I-RR, further improving robustness evaluation. The primary motivation for using I-RR and I-CREI in ROME is to overcome the limitations of RR and CREI, ensuring a more reliable and fair assessment of adversarial robustness, especially in cases where a single attack method dominates. These distinctions have been added to the manuscript for better clarity.
>
> ---
>
> - **Q2: Does the proposed method require a longer time for executing, comparing with baseline methods? If so, which part consumes the most resources?**
> - **R2:**
> Thank you for your question. ROME does introduce **additional computational overhead** compared to standard dataset distillation methods, mainly due to the robust prior generation. As shown in **Table 1**, ROME requires more training time than other methods. However, it does not necessitate full **adversarial retraining**, which is typically much more computationally expensive. The primary computational cost stems from the **adversarial perturbation** step used to generate the **robust priors**. Despite ROME having higher training time, its total time is lower compared to methods requiring adversarial retraining. We have included a detailed analysis of the computational costs compared to baseline methods in the supplementary material.
>
>     **Table 1: Comparison of Training Time for ROME and Adversarially Distilled Datasets Using DD Methods under CIFAR-10 IPC-50.**
>     | Method | Training Time (hrs) | +Adversarial Training (hrs) | Total (hrs) |
>     |--------|---------------------|-----------------------------|-------------|
>     | DC     | 1.009               | 2.019                       | 3.028       |
>     | DSA    | 0.898               | 1.796                       | 2.694       |
>     | MTT    | 0.882               | 1.764                       | 2.647       |
>     | DM     | 0.963               | 1.925                       | 2.888       |
>     | IDM    | 0.895               | 1.790                       | 2.685       |
>     | BACON  | 0.874               | 1.748                       | 2.622       |
>     | **ROME**   | **1.014**               | **1.014**                       | **2.027**       |
>
> ---
>
> - **Q3: As shown in Figure 4, looks like the robustness-aligned term has a higher impact on model performance, comparing with performance-aligned term. Any thoughts on it?**
> - **R3:**
> Thank you for your question. The observed difference in impact between the **robustness-aligned** and **performance-aligned terms** can be attributed to their distinct roles in model optimization. According to *Eq. 12*, $\alpha$ controls the trade-off between these two terms, where a higher $\alpha$ increases the weight of the robustness-aligned term, thereby enhancing the model's ability to generalize under adversarial perturbations. However, as shown in *Figure 4*, while emphasizing robustness improves performance under adversarial attacks, an excessive focus on robustness (i.e., a high $\alpha$) can suppress **non-robust features** critical for classification, potentially **reducing overall accuracy**. This trade-off is further emphasized in Table 3, where the robustness-aligned term (AP) contributes more significantly to **improving robustness** than the performance-aligned term (RPM). We have updated the discussion to clarify this trade-off and its impact on model performance.

---

### Decision · Program_Chairs · 2025-05-01

**Decision:**

Accept (poster)

**Comment:**

The paper presents a dataset distillation approach that, by incorporating a information bottleneck penalty, enhances adversarial robustness. The latter is validated both through theoretical guarantees as well as experiments.

Reviewers overall appreciated the contribution. Both the detailed performance analysis w.r.t. training time that was reported in the rebuttal, as well as the adversarial DD benchmarks should absolutely be added to the paper. Additional feedback by the reviewers (e.g., in the clarification around Eq. (111)) should also be incorporated.